# Minimax Optimal Kernel Operator Learning via Multilevel Training

**Jikai Jin**
School of Mathematical Sciences
Peking University
Beijing, China
`jkjin@pku.edu.cn`

**Yiping Lu**
Institute for Computational & Mathematical Engineering
Stanford University
Stanford, CA, US
`yplu@stanford.edu`

**Jose Blanchet**
Management Science and Engineering
Stanford University
Stanford, CA, US
`jose.blanchet@stanford.edu`

**Lexing Ying**
Department of Mathematics
Stanford University
Stanford, CA, US
`lexing@stanford.edu`

## Abstract

Learning mappings between infinite-dimensional function spaces have achieved empirical success in many disciplines of machine learning, including generative modeling, functional data analysis, causal inference, and multi-agent reinforcement learning. In this paper, we study the statistical limit of learning a Hilbert-Schmidt operator between two infinite-dimensional Sobolev reproducing kernel Hilbert spaces (RKHSs). We establish the information-theoretic lower bound in terms of the Sobolev Hilbert-Schmidt norm and show that a regularization that learns the spectral components below the bias contour and ignores the ones above the variance contour can achieve the optimal learning rate. At the same time, the spectral components between the bias and variance contours give us flexibility in designing computationally feasible machine learning algorithms. Based on this observation, we develop a multilevel kernel operator learning algorithm that is optimal when learning linear operators between infinite-dimensional function spaces.

## 1 Introduction

Supervised learning of operators between two infinite-dimensional spaces has attracted attention in several areas of application of machine learning, including, scientific computing (Lu et al., 2019; Li et al., 2020; de Hoop et al., 2021; Li et al., 2018; 2021b), functional data analysis (Crambes & Mas, 2013; Hörmann & Kidziński, 2015; Wang et al., 2020a), mean-field games (Guo et al., 2019; Wang et al., 2020b), conditional kernel mean embedding (Song et al., 2009; 2013; Muandet et al., 2017) and econometrics (Singh et al., 2019; Muandet et al., 2020; Dikkala et al., 2020; Singh et al., 2020). Despite the empirical success of operator learning, the statistical limit of learning an infinite-dimensional operator has not been investigated studied. In this paper, we study the problem of learning Hilbert Schmidt operators between infinite-dimensional Sobolev RKHSs $\mathcal{H}_K^\beta$ and $\mathcal{H}_L^\gamma$ with given kernels $k$ and $l$, respectively with $\beta, \gamma \in [0, 1)$ (Adams & Fournier, 2003; Christmann & Steinwart, 2008; Fischer & Steinwart, 2020). Our goal is to derive the optimal sample complexity for linear operator learning, *i.e.* how much data is required to achieve a certain performance level.

We first establish an information-theoretic lower bound for learning a Hilbert-Schmidt operator between Sobolev spaces with respect to a general Sobolev norm. Our information-theoretic lower bound indicates that the optimal learning rate is determined by the minimum of two polynomial rates: one is purely decided by the input Sobolev reproducing kernel Hilbert space and its evaluating norm, while the other one is purely determined by the output space along with its evaluating norm. The rate is novel in that all existing results (Fischer & Steinwart, 2020; Li et al., 2022; de Hoop et al., 2021) only establish rates that depend on the parameter of input space. The reason is all previous

works (Talwai et al., 2022; Li et al., 2022; de Hoop et al., 2021) only consider the case of the output space as a subspace of a trace bounded reproducing kernel Hilbert space but not a general Sobolev space. We refer to Remark 2.1 for detailed comparisons.

To design a learning algorithm for approximating an infinite-dimensional operator, we need to learn a finite-dimensional restriction instead of the whole operator, as the latter would result in infinite variance. The finite-dimensional selection leads to bias error but decreases the variance. A natural task is then to study the shape of regularization that can lead to the optimal bias-variance trade-off and achieve the optimal learning rate. In this paper, we consider the bias and variance contour at the scale of optimal learning. Once the regularization enables one to learn all the spectral parts above the bias contour and below the variance contour, the learning is optimal. Finally, utilizing the region between the bias contour and variance contour, we developed a multilevel training algorithm (Lye et al., 2021; Li et al., 2021a) which first learns the mapping on low frequency and then successively fine-tunes the machine learning models to fit the high-frequency output. The intuition of our algorithm aligns with the original motivation of multilevel Monte Carlo (Giles, 2008; 2015): we use the next level to reduce bias while keeping the variance at the same scale. We demonstrate that such a multilevel algorithm can achieve an optimal non-parametric rate for linear operator learning.

## 1.1 RELATED WORK

**Machine Learning Based PDE Solver**   Solving partial differential equations (PDEs) plays a prominent role in many scientific and engineering disciplines, such as physics, chemistry, operation management, macro-economy, etc. The recent deep learning breakthrough has drawn attention to solving PDEs via machine learning methods (Raissi et al., 2019; Han et al., 2018; Sirignano & Spiliopoulos, 2018; Yu et al., 2018; Khoo et al., 2019; Chen et al., 2021). The statistical power and computational cost of these problem is well-studied by recent papers (Lu et al., 2021; 2022; Nickl et al., 2020; Nickl & Wang, 2020). This paper focuses on operator learning (Chen & Chen, 1995; Long et al., 2018; 2019; Feliu-Faba et al., 2020; Khoo et al., 2021; Lu et al., 2019; Li et al., 2020; Kovachki et al., 2021; Stepaniants, 2021), *i.e.* learning a map between two infinite-dimensional function spaces. For example, one can learn a PDE solver that maps from the boundary condition to the solution or an inverse problem that maps from the boundary measurement to the coefficient field. Regarding the mathematical foundation of operator learning, (Liu et al., 2022) considers the learning rate of non-parametric operator learning. However, non-parametric functional data analysis often suffers from slower-than-polynomial convergence rates (Mas, 2012) due to the small ball probability problem for the probability distributions in infinite dimensional spaces (Delaigle & Hall, 2010). The most relevant works are (Lin et al., 2011; Reimherr, 2015; de Hoop et al., 2021), which consider the rates for learning a linear operator. For the comparison between our work and (de Hoop et al., 2021), see Remark 2.1.

**Learning with Kernel.**   Supervised least square regression in RKHS and its generalization capability have been thoroughly studied (Caponnetto & De Vito, 2007; Smale & Zhou, 2007; De Vito et al., 2005; Rosasco et al., 2010; Mendelson & Neeman, 2010). The minimax optimality with respect to the Sobolev norm has been discussed recently in (Fischer & Steinwart, 2020; Liu & Li, 2020; Lu et al., 2022). Our paper is highly related to recent works (Schuster et al., 2020; Mollenhauer & Koltai, 2020; Talwai et al., 2022; Li et al., 2022; Park & Muandet, 2020; Singh et al., 2019; 2020) on identifying the Sobolev norm learning rate for the kernel mean embedding(Song et al., 2009; 2013; Muandet et al., 2017), which can also formulated as learning an operator. The difference between our work and (Talwai et al., 2022; Li et al., 2022) sees Remark 2.1. A concurrent paper (Balasubramanian et al., 2022) considers a unified RKHS methodology for functional data analysis. Our paper provided a refined analysis and provided information theortical optimal rates for this problem.

**Multilevel Monte Carlo**   By combining biased estimators with multiple stepsizes, multilevel Monte Carlo (MLMC) (Giles, 2008; 2015) dramatically improves the rate of convergence and achieves in many settings the canonical square root convergence rate associated with unbiased Monte Carlo (Rhee & Glynn, 2015; Blanchet & Glynn, 2015). Multilevel Monte Carlo can also be used for a random variable with infinite variance (Blanchet & Liu, 2016; Chen et al., 2020). To the best of our knowledge, this is the first paper that provides optimal sample complexity for multilevel Monte Carlo type algorithm for infinite variance problems in the non-parametric regime. Very recently,

(Lye et al., 2021; Li et al., 2021a) developed a multilevel machine learning Monte Carlo algorithm (ML2MC) / multilevel fine-tuning algorithm for learning solution maps by first learning the map on the coarsest grid and then successively fine-tuning the network on samples generated at finer grids. The authors also showed that, following the telescoping in MLMC, the multilevel training procedure could reduce the generalization error without spending more time on generating training samples. (Schäfer & Owhadi, 2021; Boullé et al., 2022) consider such multi-scale algorithm for learning Green's function. However, the statistical power of such an algorithm is still under investigation. Another difference with (Boullé et al., 2022) is that we consider the Green function in $H^{-1}$ norm rather than the $\ell_1$ norm used in (Boullé et al., 2022). In this paper, we qualify a specific setting where this multilevel procedure can and is necessary to achieve the minimax optimal learning rate.

## 1.2 CONTRIBUTION

- We derive a novel information-theoretic lower bound of learning a linear operator between two infinite-dimensional Sobolev reproducing kernel Hilbert spaces. The optimal learning rate is a minimum of two polynomial rates, one only dependent on the parameters of the input space while the other only on the parameters of the output space. The first rate aligns with the previous works (Li et al., 2022), while the second is novel to the literature.

- We study the shape of regularization that can lead to the optimal learning rate. One should learn all the spectral parts under the bias contour at the level of the optimal learning rate but not the spectral components above the variance contour at the level of the learning rate. This enables the estimator to enjoy an optimal balance of bias-variance.

- We qualify a specific setting where a multilevel training procedure (Lye et al., 2021; Li et al., 2021a) is necessary and capable of achieving a minimax optimal learning rate for learning a linear operator. We perform the optimal learning rate via $O(\ln \ln n)$ ensemble of ridge regression models. This differs from finite-dimensional operator learning, where a single-level estimator can be optimal.

## 2 PROBLEM FORMULATION

### 2.1 PRELIMINARY

Let $P_K$ be a distribution over the input space $\mathcal{H}_K$ and define covariance operator $\mathcal{C}_{KK} = \mathbb{E}_{u \sim P_K} u \otimes u$. Consider its spectral decomposition $\mathcal{C}_{KK} = \sum_{i=1}^{+\infty} \mu_i^2 e_i \otimes e_i$, where $\{\mu_i^{\frac{1}{2}} e_i\}_{i=1}^{+\infty}$ is an orthogonal eigenbasis and $\{\mu_i\}$ is the corresponding eigenvalues of $\mathcal{C}_{KK}$ (here the $g \otimes h$ is an operator defined as $g \otimes h = gh^* : f \to \langle f, h \rangle g$). In the typical machine learning applications, the test distribution is the same as the training distribution, so we can assume that $\mathcal{H}_K = \left\{ \sum_i a_i \mu_i^{\frac{1}{2}} e_i : \{a_i\}_{i=1}^{\infty} \in \ell_2 \right\}$ without loss of generality. Note that this automatically holds in the context of learning the conditional mean embedding (CME) (Fischer & Steinwart, 2020; Talwai et al., 2022; Li et al., 2022).

Following Christmann & Steinwart (2008); Fischer & Steinwart (2020), we define the interpolation Sobolev space $\mathcal{H}_K^{\beta} = \left\{ f = \sum_i a_i (\mu_i^{\frac{\beta}{2}} e_i) : \{a_i\}_{i=1}^{\infty} \in l^2 \right\}$ for any $\beta > 0$, equipped with Sobolev norm defined by the inner product $\left\langle \sum_i a_i (\mu_i^{\beta/2} e_i), \sum_i b_i (\mu_i^{\beta/2} e_i) \right\rangle_{\mathcal{H}_K^{\beta}} = \sum_i a_i b_i$. For the output space, we fix a user-specified distribution $Q_L$ and a reproducing Kernel Hilbert Space. We can similarly define the covariance operator $\mathcal{C}_{Q_L}$ and the Sobolev space $\mathcal{H}_L^{\gamma}$. Natural choices of $Q_L$ include some distribution on kernel functions $\{\ell(y, \cdot) : y \in Y\}$ of $\mathcal{H}_L$ induced by some distribution $Q_L$ on $Y$, so that $\mathcal{C}_{Q_L}$ is a kernel integral operator with respect to $Q_L$ and $\mathcal{H}_L^{\gamma}$ is an interpolation space between $\mathcal{H}_L$ and $\mathcal{L}^2(Q_L)$; see Example 2.1 for a specific example.

Following Li et al. (2022), in this paper we consider the Hilbert-Schmidt norm between two Sobolev Spaces for all the operators, which is defined as following.

**Definition 2.1** (($\beta, \gamma$)-**norm**) *Let $T : \mathcal{H}_K \mapsto \mathcal{H}_L$ be a possibly unbounded linear operator. $I_{1,\beta,P_K} : \mathcal{H}_K \mapsto \mathcal{H}_K^{\beta}, \beta \in (0, 1)$ is the canonical embedding mapping that takes $u \in \mathcal{H}_K$ to the same element $u$ in the larger space $\mathcal{H}_K^{\beta}$, and $I_{1,\gamma,Q_L} : \mathcal{H}_L \mapsto \mathcal{H}_L^{\gamma}, \gamma < 1$ is similarly defined.*

*Then the $(\beta, \gamma)$-norm of $T$ is defined as*

$$\|T\|_{\beta,\gamma} = \left\|(I_{1,\gamma,Q_L}^*)^\dagger \circ T \circ I_{1,\beta,P_K}^*\right\|_{\mathrm{HS}(\mathcal{H}_K^\beta, \mathcal{H}_L^\gamma)} = \left\|\mathcal{C}_{Q_L}^{-(1-\gamma)/2} \circ T \circ \mathcal{C}_{KK}^{(1-\beta)/2}\right\|_{\mathrm{HS}(\mathcal{H}_K, \mathcal{H}_L)},$$

*where we omit the dependence of $\|\cdot\|_{\beta,\gamma}$ on $P_K$ and $Q_L$ since it will always be clear from context.*

## 2.2 PROBLEM FORMULATION

We consider the problem of learning an unknown linear operator $\mathcal{A}_0 : \mathcal{H}_K \mapsto \mathcal{H}_L$ between two reproducing kernel Hilbert spaces corresponding to kernl $k$ and $l$ respectively. We are given $N$ noisy data pairs $(u_i, v_i), 1 \leqslant i \leqslant N$ related by

$$v_i = \mathcal{A}_0 u_i + \varepsilon_i \tag{1}$$

where $u_i \overset{\text{i.i.d.}}{\sim} P_K$ for some unknown distribution $P_K$ and $\varepsilon_i$ is the noise drawn from some distribution with zero mean that may depend on $u_i$. We use $P_{KL}$ for the joint distribution of $(u_i, v_i)$. Denote $\mathcal{C}_{KK} = \mathbb{E}_{u \sim P_K} u \otimes u$, $\mathcal{C}_{KL} = \mathbb{E}_{(u,v) \sim P_{KL}} u \otimes v$ and its adjoint $\mathcal{C}_{LK} = \mathcal{C}_{KL}^*$ be uncentered cross-covariance operators associated with $P_{KL}$. Then we can reformulate the ground turth operator as $\mathcal{A}_0 = \mathcal{C}_{LK} \mathcal{C}_{KK}^\dagger$, where $\dagger$ is the pseudo-inverse (Talwai et al., 2022; Li et al., 2022). With the goal of understanding the relative difficulty of learning different types of linear operators, we investigate the sample efficiency of learning $\mathcal{A}_0$ under certain source assumptions imposed on the data model (1). Source condition (Caponnetto & De Vito, 2007; Mendelson & Neeman, 2010; Steinwart et al., 2009; Rosasco et al., 2010; Fischer & Steinwart, 2020) assumes that the learning target lies in a parameterized function class and study the learning rate for different problems with different hardness. Specifically, the source condition assume that the learning target is bounded in certain Sobolev norm. In this paper, we consider learning an operator with bounded $(\beta, \gamma)$-norm, which is the Hilbert-Schmidt norm that maps from $\mathcal{H}_K^\beta$ to $\mathcal{H}_L^\gamma$. We consider the generalization error/convergence rate under another $(\beta', \gamma')$-norm as in Fischer & Steinwart (2020); Lu et al. (2022); Talwai et al. (2022); Li et al. (2022).

**Remark 2.1** *Although recent works have considered similar problems in the context of conditional mean embedding (Talwai et al., 2022; Li et al., 2022) and functional data analysis (de Hoop et al., 2021), in all these papers, the output space is a trace bounded RKHS (de Hoop et al., 2021, Assumption 2.14 (vi)) rather than the general parameterized Sobolev space in our paper.*

We then list all the assumptions imposed on the underlying kernel for our theoretical results. We follow the standard capacity assumptions and embedding properties used in kernel regression (Fischer & Steinwart, 2020; Talwai et al., 2022; Li et al., 2022).

**Assumption 2.1 (Capacity Condition of the Covariance)** *The eigenvalues $\{\mu_i\}_{i \geqslant 1}$ of the covariance operator $\mathcal{C}_{KK} = \mathbb{E}_{u \sim P_K} u \otimes u$ satisfies $\mu_i \propto i^{-\frac{1}{p}}$ for some $p \in (0,1)$. Similarly, the eigenvalues $\{\rho_i\}_{i \geqslant 1}$ of the covariance operator $\mathcal{C}_{Q_L} = \mathbb{E}_{v \sim Q_L} v \otimes v$ satisfies $\rho_i \propto i^{-\frac{1}{q}}$ for some $q \in (0,1)$.*

**Assumption 2.2 ($\ell_\infty$ Embedding Property of the Input RKHS)** *There exists a smallest $\alpha \in (0,1)$ such that $\left\|\left(I_{1,\alpha,P_K}^*\right)^\dagger f\right\|_{\mathcal{H}_K^\alpha} \leqslant A_1$ a.s. under $P_K$ for some $A_1 < +\infty$.*

**Assumption 2.3 ($\ell_\infty$ Embedding Property of the Output RKHS)** *There exists $A_2 < +\infty$ such that $\|g\|_{\mathcal{H}_L} \leqslant A_2$ holds for all $g$ in the range of $\mathcal{A}_0$, except from a $Q_L$-null set.*

**Assumption 2.4 (Moment Condition)** *There exists an operator $V : \mathcal{H}_L \mapsto \mathcal{H}_L$ with $\mathrm{tr}\,(V) \leqslant \sigma^2$ such that for every $u \in \mathcal{H}_K$, We have $\mathbb{E}_{v \sim P_{KL}(\cdot|u)}\left[((v - \mathcal{A}_0 u) \otimes (v - \mathcal{A}_0 u))^k\right] \preceq \frac{1}{2}(2k)! R^{2k-2} V$. holds for all $k \geq 2$.*

**Assumption 2.5 (Source Condition)** *$\mathcal{A}_0$ is bounded under $(\beta, \gamma)$-norm i.e. $\|\mathcal{A}_0\|_{\beta,\gamma} \leqslant B$.*

## 2.3 EXAMPLES

In this section, we will introduce two examples of our theory. The first one is about learning a differential operator, for example inferring an advection-diffusion model Portone & Moser (2022) from observations or predicting the future Long et al. (2018); Lu et al. (2019); Li et al. (2020); Feliu-Faba et al. (2020); Huang et al. (2021). The second example is about learning conditional mean embedding Song et al. (2009; 2013); Muandet et al. (2017), which represents a conditional distribution as an RKHS element. Thus conditional distribution regression can be reduced to kernel operator learning. Our theory can also be used for linear inverse problems such as radial electrical impedance tomography (EIT) Mueller & Siltanen (2012), and the severely ill-posed inverse boundary problem for the Helmholtz equation with unknown wave-number parameter Agapiou et al. (2014). For detailed discussion, we refer to (de Hoop et al., 2021, Section 1.3)

**Example 2.1 (Learning differential operators)** *Suppose that the ground-truth operator $\mathcal{A}_0 = \Delta^t$ where $\Delta$ is the Laplacian and $t \in \mathbb{Z}$. Let $\mathcal{H}_K = \mathcal{H}^{m+2t}([0,1])$ be the Sobolev space with smoothness $m + 2t$ on $[0,1]$ and $\mathcal{H}_L = \mathcal{H}^m([0,1])$, then $\mathcal{A}_0$ is a bounded operator from $\mathcal{H}_K$ to $\mathcal{H}_L$ which corresponds to the $\beta = \gamma = 1$ case. However, we will see below that we can obtain a better characterization of the learning error using our theory.*

*Consider for example that the input has mean zero and the Matérn-type covariance operator $C_{KK} = \sigma^2 \left( -\Delta + \tau^2 I \right)^{-s}$. Its eigenvalues satisfy $\mu_n \propto n^{-2s}$. On the other hand, we choose $Q_Y$ to be a distribution supported on $\{\ell(y, \cdot) : y \in [0,1]\}$ induced by a uniform distribution on $[0,1]$, where $\ell$ is the kernel function of $\mathcal{H}_L$. Then $\mathcal{C}_{Q_Y}$ is essentially the kernel integral operator on $\mathcal{H}_L$ w.r.t. the uniform distribution, and its eigenvalues are $\rho_n \propto n^{-2m}$. The assumption $\|\mathcal{A}_0\|_{\beta,\gamma} < +\infty$ is satisfied if and only if $(1-\gamma)m < (1-\beta)s - \frac{1}{2}$ (i.e. $\gamma > 1 - \frac{2(1-\beta)s-1}{2m}$).*

**Example 2.2 (Conditional mean embedding)** *Suppose that we would like to learn the conditional distribution $P(y \mid x)$ from a data set $\{(x_i, y_i) : 1 \leqslant i \leqslant N\} \subset X \times Y$ where $x_i \overset{\text{i.i.d.}}{\sim} P_K$. Let $\mathcal{H}_K$ and $\mathcal{H}_L$ be two RKHSs on $X$ and $Y$ respectively, with measurable kernel $k(\cdot, \cdot)$ and $\ell(\cdot, \cdot)$. Then we can define a conditional mean embedding (CME) operator $C_{Y|X}$ that satisfies*

$$C_{Y|X}k(x, \cdot) = \mathbb{E}_{Y|x}\ell(Y, \cdot) =: \mu_{Y|x}, \text{ and } \mathbb{E}_{Y|x}g(Y) = \langle g, \mu_{Y|x}\rangle \, \forall x \in X.$$

*We choose $\mathcal{A}_0 = C_{Y|x}$. In this case, $\mathcal{C}_{KK} = \mathbb{E}_{P_K}k(X, \cdot) \otimes k(X, \cdot)$. Assumption 2.2 states that $\sup_{x \in X} k^\alpha(x, x) = A_1$, while Assumption 2.3 is equivalent to $\sup_{x \in X} \|\mu_{Y|x}\| \leqslant A_2$ (for simplicity we only focus on the case $\zeta = 1$). According to Assumption 2.5, we assume that $\|C_{Y|X}\|_{\beta,\gamma} \leqslant B$.*

*The mis-specified setting where $\beta < 1$ has been studied in previous work (Fischer & Steinwart, 2020; Talwai et al., 2022; Li et al., 2022). However, they only consider the case $\gamma = 1$. Our results also cover the case $\gamma < 1$, which allows us to obtain theoretical guarantee for computing conditional expectation of the larger function class $\mathcal{H}_L^\gamma$.*

## 3 INFORMATION THEORETIC LOWER BOUND

In this section, we provide an information-theoretic lower bound via the Fano method for the convergence rate of the operator learning problem formulated in Section 2.

**Theorem 3.1** *Suppose that $P_K$ and $Q_L$ are probability distributions on Hilbert spaces $\mathcal{H}_K$ and $\mathcal{H}_L$ respectively such that Assumptions 2.1 and 2.2 hold. Then for any estimator $\mathcal{L} : (\mathcal{H}_K \times \mathcal{H}_L)^{\otimes N} \mapsto \text{HS}\left(\mathcal{H}_K^\beta, \mathcal{H}_L^\gamma\right)$, there exists a linear operator $\mathcal{A}_0$ and a joint data distribution $P_{KL}$ with marginal distribution $P_K$ on $\mathcal{H}_K$ satisfying Assumptions 2.3 to 2.5, such that with probability $\geqslant 0.99$ over $(u_i, v_i) \overset{\text{i.i.d.}}{\sim} P_{KL}$ we have $\left\|\mathcal{L}\left(\{(u_i, v_i)\}_{i=1}^N\right) - \mathcal{A}_0\right\|_{\beta',\gamma'}^2 \gtrsim N^{-\min\left\{\frac{\max\{\alpha,\beta\}-\beta'}{\max\{\alpha,\beta\}+p}, \frac{\gamma'-\gamma}{1-\gamma}\right\}}$.*

**Remark 3.1** *Our lower bound is composed of a minimum of two parts. The first rate $N^{-\frac{\max\{\alpha,\beta\}-\beta'}{\max\{\alpha,\beta\}+p}}$ is the minimax optimal Sobolev learning rate for kernel regression (Fischer & Steinwart, 2020; Talwai et al., 2022; Li et al., 2022; Lu et al., 2022) and is fully determined by the parameter of the input Sobolev reproducing kernel Hilbert space. Our second rate $N^{-\frac{\gamma'-\gamma}{1-\gamma}}$ is novel to the literature.*

*This bound shows how the infinite-dimensional problem is different from the finite-dimensional regression problem and is fully determined by the output Sobolev reproducing kernel Hilbert space parameter. Our lower bound shows that the hardness of learning a linear operator is determined by the harder part between the input and output spaces. We will explain why the lower bound has such a structure in Remark 4.2 and Figure 2.*

## 4 ON THE SHAPE OF REGULARIZATION

In this section, we aim to understand the shape of regularization so that the constructed estimator $\hat{\mathcal{A}}$ based on $N$ i.i.d. data $\{(u_i, v_i)\}_{i=1}^n \sim P_{KL}^{\otimes n}$ for $1 \leqslant i \leqslant N$ enjoys an optimal learning rate.

Compared with existing approaches where a regularized least-squares estimator can achieve statistical optimality (Fischer & Steinwart, 2020; Talwai et al., 2022; Li et al., 2022; de Hoop et al., 2021) under $(\beta, 1)$-norm, we study the learning rate under the $(\beta, \gamma)$-norm ($\beta' \in (0, \beta), \gamma' \in (\gamma, 1)$) which is defined in Definition 2.1 as $\left\| \hat{\mathcal{A}} - \mathcal{A}_0 \right\|_{\beta', \gamma'} = \left\| \mathcal{C}_{Q_L}^{-\frac{1-\gamma'}{2}} \left( \hat{\mathcal{A}} - \mathcal{A}_0 \right) \mathcal{C}_{KK}^{\frac{1-\beta'}{2}} \right\|_{\mathrm{HS}(\mathcal{H}_K, \mathcal{H}_L)}$. The norm of the additional $\mathcal{C}_{Q_L}^{-\frac{1-\gamma'}{2}}$ term is unbounded which make our setting harder than the convergence in $(\beta, 1)$-norm in existing works. Since $\mathcal{C}_{Q_L}^{-\frac{1-\gamma'}{2}}$ is bounded when restricted to the finite-dimensional space $\mathrm{span}\left( \rho_i^{\frac{1}{2}} f_i : 1 \leqslant i \leqslant n \right)$, we should also include another bias-variance trade-off via regularizing in the output shape. As a result, we are interested in answering the following question

*What is the optimal way to combine the regularization in the input space and regularization in the output space? i.e. What is the optimal shape of regularization?*

To answer this question, we investigate the problem in the spectral space, *i.e.* considering the spectral representation of operator $\mathcal{A}_0 = \sum_{i,j=1}^{+\infty} a_{ij} \mu_i^{\frac{\beta}{2}} e_i \otimes \rho_j^{1-\frac{\gamma}{2}} f_j$. The problem of estimating $\mathcal{A}$ then reduces to learning the coefficients "matrix" $(a_{ij})_{i,j=1}^\infty$. The source condition Assumption 2.5 enforces $\sum_{i,j=1}^\infty a_{ij}^2 \leq B$. We show in Appendix B.1.1 that regularizing the basis $e_i \otimes f_j$ will introduce a bias of order $\left\| a_{ij} \mu_i^{\frac{\beta}{2}} e_i \otimes \rho_j^{1-\frac{\gamma}{2}} f_j \right\|_{\beta', \gamma'}^2 = a_{ij}^2 \mu_i^{\beta - \beta'} \rho_j^{\gamma' - \gamma} \propto i^{-\frac{\beta - \beta'}{p}} j^{-\frac{\gamma' - \gamma}{q}}$ under the $(\beta', \gamma')$-norm. On the other hand, when $\alpha \leqslant \beta + p$, we show in Appendix B.1.2 that the variance of learning $(i, j)$ from noisy data scales as $\frac{1}{N} \mu_i^{-\beta'} \rho_j^{-(1-\gamma')} \propto \frac{1}{N} i^{\frac{\beta'}{p}} j^{\frac{1-\gamma'}{q}}$. Since the variance would accumulate for a fixed $j$, learning $(i, j)$ for $i \leqslant i_{\max}$ results in a variance of $\propto \frac{1}{N} i_{\max}^{\frac{\beta' + p}{p}} j^{\frac{1-\gamma'}{q}}$. (Similar analysis can be carried out for the $\alpha > \beta + p$ case as well, but the variance now scales as $\frac{1}{N} i_{\max}^{\frac{\beta' + \alpha - \beta}{p}} j^{\frac{1-\gamma'}{q}}$; see Appendix B for detailed derivations.) In summary, we need to make bias-variance trade off in the $(i, j)$−plane, *i.e.* decide whether we should learn or regularize over the basis $e_i \otimes f_j$.

### 4.1 REGULARIZATION VIA VARIANCE CONTOUR

The underlying idea of regularization is that some components are intrinsically hard to learn due to large variance; these components are then neglected by adding regularization and are counted as bias. Thus, the remaining components are easy to learn due to controllable variance. This intuition works well when the estimation error results from the noise of the data and is well-studied in a line of works (Fischer & Steinwart, 2020; de Hoop et al., 2021; Talwai et al., 2022; Li et al., 2022). This idea still works in our setting, but we need to re-evaluate the bias and variance of each component. Since we work with the Hilbert-Schmidt norm, this can be done in a coordinate-wise manner, meaning that we can look at each $a_{ij}$ separately and decide whether to neglect it (contribute to bias) or to learn it from data (contribute to variance).

Since the variance term measures the hardness of learning, we naturally introduce the notion of *variance contour*, which is a curve on the $\mathbb{R}_+^2$ plane on which all points induce the same order of variance (here we work with real coordinates for convenience, although we only care about integer

 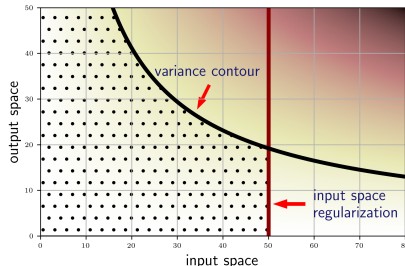

Figure 1: An illustration of our proposed regularization scheme. Left: the regularized least-squares estimator studied in previous works (Fischer & Steinwart, 2020; de Hoop et al., 2021; Talwai et al., 2022) which only regularizes on the input space. Right: our double regularization scheme via variance contour can achieve the optimal convergence rate in our setting.

points). Formally, we fix an arbitrary constant $C > 0$ and define

$$\ell_{C,\mathrm{var}} = \left\{ (x,y) \in \mathbb{R}_+^2 : x^{\frac{\beta' + \max\{\alpha - \beta, p\}}{p}} y^{\frac{1-\gamma'}{q}} = C \right\}. \tag{2}$$

A reasonable regularization scheme is then to learn all coordinates $(i,j) \in \mathbb{Z}_+^2$ *below* the curve $\ell_{C,\mathrm{var}}$ and 'regularize out' the remaining coordinates that are difficult to learn due to large variances. This can gives us the estimator with smallest estimator at give variance level. This observation motivates us to construct our estimator as

$$\hat{\mathcal{A}} = \sum_{j=1}^{y_N} \left( \rho_j^{\frac{1}{2}} f_j \otimes \rho_j^{\frac{1}{2}} f_j \right) \hat{\mathcal{C}}_{LK} \left( \hat{\mathcal{C}}_{KK} + \lambda_j I \right)^{-1}, \tag{3}$$

where $\hat{C}_{LK} = \frac{1}{N} \sum_{i=1}^{N} v_i \otimes u_i$, $\lambda_j (1 \leqslant j \leqslant y_N = C^{\frac{q}{1-\gamma'}})$ are the regularization coefficients imposed on different dimensions of the output space. According to (2) and noting that $\mu_i \propto i^{-\frac{1}{p}}$, we define

$$\lambda_j = \max \left\{ \left( j^{-\frac{1-\gamma'}{q}} N^{\max\left\{ 1 - \frac{\beta - \beta'}{\max\{\alpha, \beta + p\}}, \frac{1-\gamma'}{1-\gamma} \right\}} \right)^{-\frac{1}{\beta'+p}}, c_0 \left( \frac{N}{\log N} \right)^{-\frac{1}{\alpha}} \right\}, \tag{4}$$

with $C = N^{\max\left\{ 1 - \frac{\beta - \beta'}{\max\{\alpha, \beta + p\}}, \frac{1-\gamma'}{1-\gamma} \right\}}$ in (2). The additional $N^{-\frac{1}{\alpha}}$ term in (4) is needed for controlling the error of approximating $\mathcal{C}_{KK}$ via $\hat{\mathcal{C}}_{KK}$ (cf. Theorem D.3) which is standard in the Sobolev learning literature Fischer & Steinwart (2020); Talwai et al. (2022); Lu et al. (2022). The following theorem describes the convergence rate of our estimator defined by (3) and (4).

**Theorem 4.1** *Consider the estimator $\hat{\mathcal{A}}$ defined by (3) and (4). Suppose that Assumptions 2.1 to 2.5 hold, then there exists a universal constant $C$ such that with probability $\geqslant 1 - e^{-\tau}$, we have $\left\| \hat{\mathcal{A}} - \mathcal{A}_0 \right\|_{\beta',\gamma'}^2 \leqslant C\tau^2 \left( \frac{N}{\log N} \right)^{-\min\left\{ \frac{\beta - \beta'}{\max\{\alpha, \beta + p\}}, \frac{\gamma' - \gamma}{1-\gamma} \right\}} \log^2 N.$*

**Remark 4.1** *Compared with Theorem 3.1, our upper bound is optimal up to logarithmic factors when $\alpha \leqslant \beta$. The optimal learning rate in the $\alpha > \beta$ regime is an outstanding problem for decades, even without the additional problem-dependent parameters $\gamma, \gamma'$ (see e.g. the discussions following (Fischer & Steinwart, 2020, Theorem 2)). In this paper, we do not address this problem either.*

## 4.2 REGULARIZATION VIA BIAS CONTOUR

We have showed that if we learn all the spectral components under certain variance contour and regularize all other component can achieve optimal rate. In this section, we introduce another scheme to design the optimal estimator via learning all the spectral component under a certain bias contour. Specifically, we consider deciding the regularization strength according to the spectral elements induce a certain level of bias i.e. the *bias contour* $\ell_{C',\mathrm{bias}} = \left\{ (x,y) \in \mathbb{R}_+^2 : x^{\frac{\beta - \beta'}{p}} y^{\frac{\gamma' - \gamma}{q}} = C' \right\}.$ does not coincide with $\ell_{C,\mathrm{var}}$ for any $C'$ up to constant scaling. Thus, there exists a point

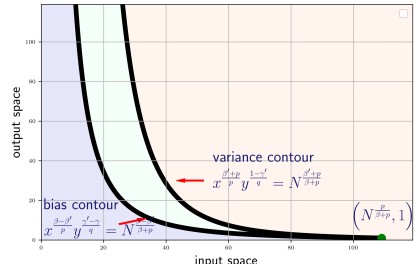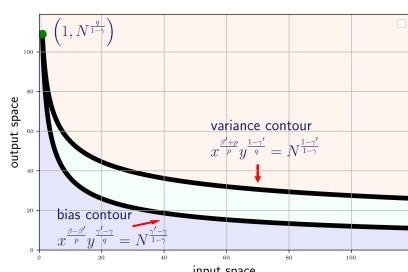

Figure 2: The plot of the bias contour and the variance contour. For simplicity, we only plot the case $\alpha \leqslant \beta + p$ here. The variance contour is always above the bias contour. Left: When $\frac{\beta'+p}{\beta+p} \geqslant \frac{1-\gamma'}{1-\gamma}$, the two yields $\mathcal{O}\left(N^{-\frac{\beta-\beta'}{\max\{\alpha,\beta+p\}}}\right)$ convergence rate. It is the same learning rate as the two kernel regression curves meet when $y = 1$. Right: When $\frac{\beta'+p}{\beta+p} \geqslant \frac{1-\gamma'}{1-\gamma}$, the two contours yield the same regularization on the output space leading to a convergence rate of $\mathcal{O}\left(N^{-\frac{\gamma'-\gamma}{1-\gamma}}\right)$.

$(x^*, y^*)$ on the variance contour with maximal contribution to bias. Naturally, we can also construct our estimator using a bias contour that passes through $(x^*, y^*)$. In this case, we may define

$$\lambda_j = \max\left\{\left(j^{-\frac{\gamma'-\gamma}{q}} N^{\min\left\{\frac{\beta-\beta'}{\max\{\alpha,\beta+p\}}, \frac{\gamma'-\gamma}{1-\gamma}\right\}}\right)^{-\frac{1}{\beta-\beta'}}, c_0\left(\frac{N}{\log N}\right)^{-\frac{1}{\alpha}}\right\}$$ for similar reasons as Section 4.1, which also yields optimal rate as stated in Theorem 4.2 below.

**Remark 4.2 (On the optimal shape of regularization)** *The discussion in Sections 4.1 and 4.2 reveals another understanding of our information theoretic lower bound. Firstly, we should learn all the spectral components under the bias contour otherwise the bias will exceed the lower bound. Secondly, we should not learn any spectral component over the variance contour since otherwise the variance will exceed the lower bound. Thus the bias contour should always be under the variance contour, otherwise no estimator can be designed. The bias and variance contours at the level of optimal learning rate are plotted in Figure 2. They only meet at $(x^*, y^*)$ with $x^* = 1$ or $y^* = 1$, which has the largest contribution to the bias (resp. variance) among all points on the variance (resp. bias) contour, thus dominating the estimation error. When the two curves meet at $y^* = 1$, it reduces to the original kernel regression case. When the two curves meet at $x^* = 1$, it leads to our new rate that depends on the output space.*

**Theorem 4.2** *Consider the estimator $\hat{\mathcal{A}}$ defined by (3) with $\lambda_j$ defined above. Suppose that Assumptions 2.1 to 2.5 hold, then there exists a universal constant $C$, such that $\left\|\hat{\mathcal{A}} - \mathcal{A}_0\right\|_{\beta',\gamma'}^2 \leqslant C\tau^2\left(\frac{N}{\log N}\right)^{-\min\left\{\frac{\beta-\beta'}{\max\{\alpha,\beta+p\}}, \frac{\gamma'-\gamma}{1-\gamma}\right\}}\log^2 N$ holds with probability $\geqslant 1 - e^{-\tau}$.*

## 5 MULTILEVEL KERNEL OPERATOR LEARNING

In this section, we study a multilevel machine learning algorithm Lye et al. (2021); Li et al. (2021a); Boullé et al. (2022) but at each level we consider a cost-accuracy trade-off De Hoop et al. (2022) to control the variance at a proper scale. We show that the multilevel level algorithm can cover all the spectral component below the bias contour and achieve the optimal learning rate. Our idea is similar to the multilevel Monte Carlo Giles (2008; 2015), which reduces bias from multilevel algorithm. Our multilevel estimator differs from the DeepONet (Lu et al., 2019) and the PCA-Net (Bhattacharya et al., 2020) since we add different regularizations for each level. Our theory indicates that the multilevel approach outperforms previous ones and achieves the optimal learning rate.

The basic idea is to design a minimum number of machine learning estimators that cover all the spectral elements under the bias contour but do not exceed the variance contour at the same time. To achieve this, we choose sequences $\{x_i\}$ and $\{y_i\}$ for $1 \leqslant i \leqslant L_N$ where $y_i$ denotes the $i$-th level and $x_i$ controls the corresponding regularization via the regularization coefficient $\lambda_i^{(K)} = x_i^{-\frac{1}{p}}$. The sequences are chosen in a staircase manner as plotted in Figure 3 (for formal definitions see

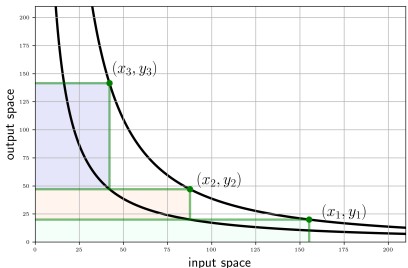 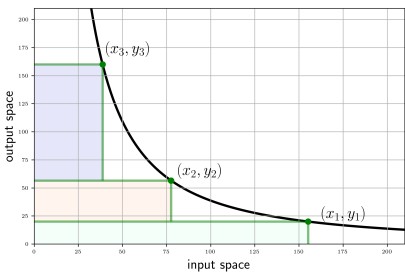

Figure 3: Construction of the sequence $\{(x_i, y_i)\}$. Left: the case $\frac{\beta-\beta'}{\max\{\alpha,\beta+p\}} \neq \frac{\gamma'-\gamma}{1-\gamma}$. Right: the case $\frac{\beta-\beta'}{\max\{\alpha,\beta+p\}} = \frac{\gamma'-\gamma}{1-\gamma}$, where the bias and variance contours overlap and we set $x_{n+1} = \frac{1}{2}x_n$. Each rectangular represents a certain level of regularization.

Appendix C). The eigenbasis $\left\{ \rho_j^{\frac{1}{2}} f_j \right\}$ of the output space is divided into defferent levels by $\{y_i\}$. The main idea behind our multilevel method is that different levels of the output need to be learned with different regularization. Formally, we define our multilevel estimator as

$$\hat{\mathcal{A}}_{\mathtt{ml}} = \sum_{i=0}^{L_N} \left( \sum_{y_{i-1} \leqslant j < y_i} \rho_j^{\frac{1}{2}} f_j \otimes \rho_j^{\frac{1}{2}} f_j \right) \hat{\mathcal{C}}_{LK} \left( \hat{\mathcal{C}}_{KK} + \lambda_i^{(K)} I \right)^{-1}. \tag{5}$$

The following theorem shows that the estimator (5) can achieve the optimal convergence rate with $L_N = \mathcal{O}(\ln \ln N)$ when $\frac{\beta-\beta'}{\max\{\alpha,\beta+p\}} \neq \frac{\gamma'-\gamma}{1-\gamma}$. We also show that $O(\ln N)$ estimator is needed for the case when $\frac{\beta-\beta'}{\max\{\alpha,\beta+p\}} = \frac{\gamma'-\gamma}{1-\gamma}$ (Figure 3 Right) in Appendix C.

**Theorem 5.1** *Suppose that Assumptions 2.1 to 2.5 hold, then there exists a sequence $\{y_i\}_{1 \leqslant i \leqslant L_N}$ with $L_N = \mathcal{O}(\ln N)$ when $\frac{\beta-\beta'}{\max\{\alpha,\beta+p\}} = \frac{\gamma'-\gamma}{1-\gamma}$ and $\mathcal{O}(\ln \ln N)$ otherwise, such that the estimator $\hat{\mathcal{A}}_{\mathtt{ml}}$ satisfies $\left\| \hat{\mathcal{A}}_{\mathtt{ml}} - \mathcal{A}_0 \right\|_{\beta',\gamma'}^2 \leqslant C\tau^2 \left( \frac{N}{\log N} \right)^{-\min\left\{ \frac{\beta-\beta'}{\max\{\alpha,\beta+p\}}, \frac{\gamma'-\gamma}{1-\gamma} \right\}} \log^2 N$ with probability $\geqslant 1 - e^{-\tau}$, where $C$ is a universal constant.*

**Remark 5.1** *Our multilevel algorithm first apply the regression algorithm on low-frequency projections of the output samples with small regularization and then successively fine-tune the regression model on high-frequency projections of the output samples with stronger regularization, which matches the empirical use (Li et al., 2021a; Lye et al., 2021).*

## 6 CONCLUSION AND DISCUSSION

We considered the sample complexity of learning an operator between two infinite-dimensional Sobolev kernel Hilbert spaces. We provided an information theoretical lower bound for this problem along with a multi-level machine learning algorithm. Our lower bound is determined by the harder rate of two polynomial rates: one is fully determined by the hardness of the input space, while the other is fully controlled by the hardness of the output space. The second rate is new to the literature. We explained our bound from the viewpoint of variance and bias counters in Remark 4.2 and Figure 2. The optimal estimator should learn all the spectral elements under the bias contour but learn no information above the variance contour. To meet this requirement, we combined the idea of multi-level Monte Carlo with kernel operator learning, using successive levels to fit higher frequency information while keeping the variance at the same scale to reduce the bias. Our paper is the first on the non-parametric statistical optimality for multi-level algorithms. We leave estimation from discretely observed functional covariates with noise as future work (Zhou et al.; 2022).

### ACKNOWLEDGMENTS

Jikai Jin is partially supported by the elite undergraduate training program of School of Mathematical Sciences in Peking University. Yiping Lu is supported by the Stanford Interdisciplinary Graduate Fellowship (SIGF). Jose Blanchet is supported in part by the Air Force Office of Scientific Research under award number FA9550-20-1-0397. Lexing Ying is supported is supported by National Science Foundation under award DMS-2208163.

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

# A    PROOF OF THE LOWER BOUND

In this section, we follow the lower bound proof in Fischer & Steinwart (2020) to give a lower bound of the convergence rate in our operator learning setting.

## A.1    PRELIMINARIES ON TOOLS FOR LOWER BOUNDS

In this section, we repeat the standard tools we use to establish the lower bound. The main tool we use is the Fano's inequality and the Varshamov-Gilber Lemma.

**Lemma A.1 (Fano's methods)** *Assume that $V$ is a uniform random variable over set $\mathcal{V}$, then for any Markov chain $V \to X \to \hat{V}$, we always have*

$$\mathcal{P}(\hat{V} \neq V) \geq 1 - \frac{I(V; X) + \log 2}{\log(|\mathcal{V}|)}$$

In our proof we will use a version from Fischer & Steinwart (2020).

**Lemma A.2** *(Fischer & Steinwart, 2020, Theorem 20) Let $M \geqslant 2$, $(\Omega, \mathcal{A})$ be a measurable space, $P_0, P_1, \ldots, P_M$ be probability measures on $(\Omega, \mathcal{A})$ with $P_j \ll P_0$ for all $j = 1, \ldots, M$, and $0 < \alpha_* < \infty$ with*

$$\frac{1}{M} \sum_{j=1}^{M} KL(P_j || P_0) \leqslant \alpha_*.$$

*Then, for all measurable functions $\Psi : \Omega \to \{0, 1, \ldots, M\}$, the following bound is satisfied*

$$\max_{j=0,1,\ldots,M} P_j(\omega \in \Omega : \Psi(\omega) \neq j) \geqslant \frac{\sqrt{M}}{1 + \sqrt{M}} \left( 1 - \frac{3\alpha_*}{\log(M)} - \frac{1}{2\log(M)} \right).$$

**Lemma A.3 (Varshamov-Gillbert Lemma, Tsybakov (2008) Theorem 2.9)** *Let $D \geq 8$. There exists a subset $\mathcal{V} = \{\tau^{(0)}, \cdots, \tau^{(2^{D/8})}\}$ of $D-$dimensional hypercube $\mathcal{H}^D = \{0, 1\}^D$ such that $\tau^{(0)} = (0, 0, \cdots, 0)$ and the $\ell_1$ distance between every two elements is larger than $\frac{D}{8}$*

$$\sum_{l=1}^{D} \|\tau^{(j)} - \tau^{(k)}\|_{\ell_1} \geq \frac{D}{8}, \text{for all } 0 \leq j, k \leq 2^{D/8}$$

## A.2    PROOF OF THE LOWER BOUND

To prove our lower bound, we construct a sequence of linear operators as follows:

$$\mathcal{A}_\omega = \sqrt{\frac{32\varepsilon}{m_1 K}} \sum_{i=1}^{m_1} \sum_{j=1}^{K} \omega_{ij} \mu_{i+m_1}^{\beta'/2} \rho_{j+m_2}^{1-\gamma'/2} f_{j+m_2} \otimes e_{i+m_1}, \quad \omega_{ij} \in \{0, 1\}$$

where $m_1$ and $m_2$ are hyper-parameters (scale as $\text{poly}(N)$ and will be selected later) and $K$ is a constant that will be specified afterwards. It's easy to check that

$$\|\mathcal{A}_\omega - \mathcal{A}_{\omega'}\|_{\beta',\gamma'}^2 \leqslant \frac{32\varepsilon}{m_1 K} \sum_{i=1}^{m_1} \sum_{j=1}^{K} \left( \omega_{ij} - \omega_{ij}' \right)^2$$

By Gilbert-Varshamov Lemma it is possible to select $M_\varepsilon \geqslant 2^{m_1 K/8}$ binary strings

$$\omega^{(1)}, \omega^{(2)}, \cdots, \omega^{(M_\varepsilon)} \in \{0, 1\}^{m_1 K}$$

such that $\left\| \omega^{(i)} - \omega^{(j)} \right\|_2^2 \geqslant 4\varepsilon$. Let $\Omega$ be the collection of this strings.

We now select the hyper-parameters to satisfies the assumptions made in Section 2. First we have

$$\|\mathcal{A}_\omega\|_{\beta,\gamma}^2 \leqslant \frac{32\varepsilon}{m_1 K} \sum_{i=1}^{m_1} \sum_{j=1}^{K} \mu_{i+m_1}^{-(\beta-\beta')} \rho_{j+m_2}^{-(\gamma'-\gamma)} \lesssim \varepsilon (2m_1)^{\frac{\beta-\beta'}{p}} (2m_2)^{\frac{\gamma'-\gamma}{q}}$$

where the last step follows from Assumption 2.1. Similarly, we have $\|\mathcal{A}_\omega\|_{\alpha,1}^2 \lesssim \varepsilon (2m_1)^{\frac{\alpha-\beta'}{p}} (2m_2)^{\frac{\gamma'-1}{q}}$. To the assumptions made in Section 2, we should make

$$(2m_1)^{\frac{\max\{\alpha,\beta\}-\beta'}{p}} (2m_2)^{\frac{\gamma'-\gamma}{q}} \lesssim \varepsilon^{-1} \tag{6}$$

be satisfied. To be specific, with the previous selection of hyper-parameters, we can have $\|\mathcal{A}_\omega\|_{\beta,\gamma} = \mathcal{O}(1)$ and

$$\sup_{g\in\text{range}(\mathcal{A}_\omega)} \|g\|_{\mathcal{H}_L} \leqslant \sup_f \|\mathcal{A}_\omega\|_{\alpha 1} \cdot \left\| \left(I_{1,\alpha,P_K}^*\right)^\dagger f \right\|_{\mathcal{H}_K^\alpha} < +\infty$$

where the last step follows from our assumption on the input distribution Assumption 2.2. This verifies that Assumptions 2.3 and 2.5 hold for $\mathcal{A}_\omega, \forall \omega \in \Omega$.

We now construct the hypothesis (probability distributions) as follows: for $\forall \omega \in \{0,1\}^{m_1}$, define

$$P_\omega(\mathrm{d}f, \mathrm{d}g) = \mathrm{d}\mathcal{N}\left(C_\omega f, \Sigma\right)(g) \cdot \mathrm{d}P_K(f)$$

where the covariance operator $\Sigma = \frac{\sigma^2}{K} \sum_{j=1}^K \rho_{j+m_2} f_{j+m_2} \otimes f_{j+m_2}$ for some constant $\sigma > 0$. It's then easy to see that $\text{tr}(\Sigma) = \sigma^2$, which satisfies Assumption 2.4 .Note that the range of $\mathcal{A}_\omega$ is $\text{span}(f_{m_2})$ and $\Sigma$ is non-degenerate on this subspace. As a result, we can view $P_\omega, \omega \in \Omega$ as distributions on $\mathcal{H}_K \times \text{span}(f_{j+m_2} : 1 \leqslant j \leqslant K)$, and we have for $\forall \omega, \omega' \in \Omega$ that

$$\begin{aligned}
KL\left(P_\omega\|P_{\omega'}\right) &= \mathbb{E}_{f\sim P_K}\left[KL\left(P_\omega(\mathrm{d}g \mid f)\|P_{\omega'}(\mathrm{d}g \mid f)\right)\right] \\
&= \mathbb{E}_{f\sim P_K}\left[KL\left(\mathcal{N}(C_\omega f, \Sigma)\|\mathcal{N}(C_{\omega'}f, \Sigma)\right)\right] \\
&= \mathbb{E}_{f\sim P_K}\left\langle(\mathcal{A}_\omega - \mathcal{A}_{\omega'})f, \Sigma^\dagger(\mathcal{A}_\omega - \mathcal{A}_{\omega'})f\right\rangle \\
&\leqslant \sigma^{-2}K\mathbb{E}_{f\sim P_K}\left\langle(\mathcal{A}_\omega - \mathcal{A}_{\omega'})f, (\mathcal{A}_\omega - \mathcal{A}_{\omega'})f\right\rangle \\
&= \frac{32\varepsilon}{m_1\sigma^2}\mathbb{E}_{f\sim P_K}\left\|\sum_{i=1}^{m_1}\sum_{j=1}^K(\omega_{ij} - \omega'_{ij})\mu_{i+m_1}^{\beta'/2}\rho_{j+m_2}^{1-\gamma'/2}\langle f, e_{i+m_1}\rangle f_{j+m_2}\right\|_{\mathcal{H}_L}^2 \\
&= \frac{32\varepsilon}{m_1\sigma^2}\mathbb{E}_{f\sim P_K}\sum_{j=1}^K\rho_{j+m_2}^{1-\gamma'}\left(\sum_{i=1}^{m_1}(\omega_{ij} - \omega'_{ij})\mu_{i+m_1}^{\beta'/2}\langle f, e_{i+m_1}\rangle\right)^2 \\
&= \frac{32\varepsilon}{m_1\sigma^2}\sum_{i=1}^{m_1}\sum_{j=1}^K(\omega_{ij} - \omega'_{ij})^2\mu_{i+m_1}^{\beta'}\rho_{j+m_2}^{1-\gamma'} \lesssim \varepsilon\sigma^{-2}m_1^{-\frac{\beta'}{p}}m_2^{-\frac{1-\gamma'}{q}}
\end{aligned}$$

where the last step follows from $\mathbb{E}_{P_K}u\otimes u = \mathcal{C}_{KK} = \sum_{i=1}^\infty \mu_i^2 e_i\otimes e_i$ and recall that $K$ is a constant. Hence we deduce that

$$\frac{1}{M_\varepsilon}\sum_{\omega'\in\Omega}KL(P_{\omega'}^n\|P_\omega^n) \lesssim \sigma^{-2}n\varepsilon m_1^{-\frac{\beta'}{p}}m_2^{-\frac{1-\gamma'}{q}} =: \alpha^*$$

Applying Lemma A.2, we find that when

$$\alpha^* \lesssim \log M_\varepsilon \Leftrightarrow \varepsilon \lesssim n^{-1}m_1^{\frac{\beta'}{p}}m_2^{\frac{1-\gamma'}{q}},$$

there exists a hypothesis $P_{\omega_0}$ such that for any estimator $\hat{\mathcal{A}}_{\omega_0}$,

$$\left\{\left\|\hat{\mathcal{A}}_{\omega_0} - \mathcal{A}_{\omega_0}\right\|_{\beta',\gamma'}^2 \gtrsim \varepsilon\right\} \supset \left\{\omega_0 \neq \arg\min_{\omega\in\Omega}\|\mathcal{A}_\omega - \mathcal{A}_{\omega_0}\|_{\beta',\gamma'}\right\}$$

holds with high probability.

Finally, we need to choose optimal $m_1$ and $m_2$ under the constraint (6). It turns out that either $m_1 = 1$ or $m_2 = 1$, and the resulting lower bound is

$$\left\|\hat{\mathcal{A}} - \mathcal{A}\right\|_{\beta',\gamma'} \gtrsim n^{-\min\left\{\frac{\max\{\alpha,\beta\}-\beta'}{2(\max\{\alpha,\beta\}+p)}, \frac{\gamma'-\gamma}{2(1-\gamma)}\right\}}.$$

# B  PROOF OF THE UPPER BOUND

In this section, we upper-bound the learning error of estimator (3) which defined as

$$\hat{\mathcal{A}} = \sum_{j=1}^{y_N} \left( \rho_j^{\frac{1}{2}} f_j \otimes \rho_j^{\frac{1}{2}} f_j \right) \hat{\mathcal{C}}_{LK} \left( \hat{\mathcal{C}}_{KK} + \lambda_j I \right)^{-1}, \tag{7}$$

where $\lambda_j, 1 \leqslant j \leqslant y_N = N^{\frac{q}{1-\gamma'} \max\{1 - \frac{\beta - \beta'}{\max\{\alpha, \beta + p\}}, \frac{1-\gamma'}{1-\gamma}\}}$ are regularization coefficients that we impose on different dimensions of the output space. In this section, we consider the following two ways to select regularization coefficients in Section 4:

- We regularize all spectral component below certain variance contour, *i.e.* we set regularization strength $\lambda_j = \max\left\{ \left( j^{-\frac{1-\gamma'}{q}} N^{\max\left\{1 - \frac{\beta - \beta'}{\max\{\alpha, \beta + p\}}, \frac{1-\gamma'}{1-\gamma}\right\}} \right)^{-\frac{1}{\beta' + p}}, c_0 \left( \frac{N}{\log N} \right)^{-\frac{1}{\alpha}} \right\}$ (4).

- We regularize all spectral component below certain bias contour, *i.e.* we set regularization strength $\lambda_j = \max\left\{ \left( j^{-\frac{\gamma' - \gamma}{q}} N^{\min\left\{ \frac{\beta - \beta'}{\max\{\alpha, \beta + p\}}, \frac{\gamma' - \gamma}{1-\gamma}\right\}} \right)^{-\frac{1}{\beta - \beta'}}, c_0 \left( \frac{N}{\log N} \right)^{-\frac{1}{\alpha}} \right\}$ (19).

To obtain the upper bound for our estimator, we decompose the learning error $\mathcal{E}(\hat{\mathcal{A}}) = \left\| \hat{\mathcal{A}} - \mathcal{A}_0 \right\|_{\beta', \gamma'}$ in to bias and variance via

$$\mathcal{E}(\mathcal{A}) \leqslant \underbrace{\left\| \hat{\mathcal{A}} - \mathcal{A}_\lambda \right\|_{\beta', \gamma'}}_{\text{variance term}} + \underbrace{\left\| \mathcal{A}_\lambda - \mathcal{A}_0 \right\|_{\beta', \gamma'}}_{\text{bias term}},$$

where

$$\mathcal{A}_\lambda = \sum_{j=1}^{y_N} \left( \rho_j^{\frac{1}{2}} f_j \otimes \rho_j^{\frac{1}{2}} f_j \right) \mathcal{C}_{KL} \left( \mathcal{C}_{KK} + \lambda_j I \right)^{-1}. \tag{8}$$

## B.1  REGULARIZATION VIA VARIANCE COUNTER

In the following, we separately bound the bias term and the variance term. We first assume $\alpha \leqslant \beta + p$ in Appendix B.1.1 and Appendix B.1.2, then the case $\alpha > \beta + p$ is treated in Appendix B.1.3. Finally in Appendix B.2, we establish the same convergence rate for regularization via bias contour.

### B.1.1  BIAS

**Lemma B.1** $\|\mathcal{A}_0 - \mathcal{A}_\lambda\|_{\beta', \gamma'}^2 \lesssim N^{-\min\left\{ \frac{\beta - \beta'}{\beta + p}, \frac{\gamma' - \gamma}{1-\gamma}\right\}}$.

***Proof sketch***: Since $\|\mathcal{A}_0\|_{\beta, \gamma} \leqslant B$, we can write $\mathcal{A}_0 := \sum_{i=1}^{+\infty} \sum_{j=1}^{+\infty} a_{ij} \mu_i^{\frac{\beta}{2}} \rho_j^{1 - \frac{\gamma}{2}} f_j \otimes e_i$ where the coefficient matrix $A_0 = (a_{ij})_{1 \leqslant i, j \leqslant +\infty}$ satisfies $\|A_0\|_F^2 \leqslant B^2$. The definition (8) implies that for $1 \leqslant j \leqslant y_N$ and $i \geqslant 1$ we have

$$\left\langle \rho_j^{\frac{1}{2}} f_j, \mathcal{A}_\lambda \mu_i^{\frac{1}{2}} e_i \right\rangle = \left\langle \rho_j^{\frac{1}{2}} f_j, \mathcal{C}_{KL} \left( \mathcal{C}_{KK} + \lambda_j I \right)^{-1} \mu_i^{\frac{1}{2}} e_i \right\rangle$$

$$= \left\langle \rho_j^{\frac{1}{2}} f_j, \mathcal{A}_0 \mathcal{C}_{KK} \left( \mathcal{C}_{KK} + \lambda_j I \right)^{-1} \mu_i^{\frac{1}{2}} e_i \right\rangle = \frac{\mu_i^{\frac{1+\beta}{2}}}{\mu_i + \lambda_j} \rho_j^{\frac{1-\gamma}{2}} a_{ij}.$$

The bias term can be bounded as follows:

$$
\begin{aligned}
\|\mathcal{A}_0 - \mathcal{A}_\lambda\|_{\beta',\gamma'}^2 &= \sum_{i,j=1}^{+\infty} \left\langle \rho_j^{\frac{1}{2}} f_j, \mathcal{C}_{Q_K}^{-\frac{1-\gamma'}{2}} (\mathcal{A}_0 - \mathcal{A}_\lambda) \mathcal{C}_{KK}^{\frac{1-\beta'}{2}} \mu_i^{\frac{1}{2}} e_i \right\rangle^2 \\
&= \sum_{j=1}^{y_N} \sum_{i=1}^{+\infty} \mu_i^{\beta-\beta'} \rho_j^{\gamma'-\gamma} \frac{\lambda_j^2}{(\mu_i + \lambda_j)^2} a_{ij}^2 \\
&\leqslant \sum_{j=1}^{y_N} \rho_j^{\gamma'-\gamma} \max_{i \geqslant 1} \left( \mu_i^{\beta-\beta'} \frac{\lambda_j^2}{(\mu_i + \lambda_j)^2} \right) \cdot \sum_{i=1}^{+\infty} a_{ij}^2 \\
&\lesssim \sum_{j=1}^{y_N} j^{-\frac{\gamma'-\gamma}{q}} \lambda_j^{-(\beta-\beta')} \sum_{i=1}^{+\infty} a_{ij}^2 \lesssim B^2 \max_{1 \leqslant j \leqslant y_N} j^{-\frac{\gamma'-\gamma}{q}} \lambda_j^{-(\beta-\beta')}.
\end{aligned}
\tag{9}
$$

We now prove that

$$
j^{\frac{\gamma'-\gamma}{q}} \lambda_j^{\beta-\beta'} \gtrsim N^{\min\left\{ \frac{\beta-\beta'}{\beta+p}, \frac{\gamma'-\gamma}{1-\gamma} \right\}}, \quad \forall 1 \leqslant j \leqslant y_N.
\tag{10}
$$

**Case 1.** If $\lambda_j = c_0 \left( \frac{N}{\log N} \right)^{\frac{1}{\alpha}}$, then

$$
j^{\frac{\gamma'-\gamma}{q}} \lambda_j^{\beta-\beta'} \geqslant \lambda_j^{\beta-\beta'} \gtrsim N^{\frac{\beta-\beta'}{\alpha}} \geqslant N^{\frac{\beta-\beta'}{\beta+p}}
$$

where we use $\alpha \leqslant \beta + p$ in the final step.

**Case 2.** If $\lambda_j = \left( N^{\max\left\{ \frac{\beta'+p}{\beta+p}, \frac{1-\gamma'}{1-\gamma} \right\}} j^{-\frac{1-\gamma'}{q}} \right)^{\frac{1}{\beta'+p}}$, we need to consider two sub-cases:

- If $\frac{\beta'+p}{\beta+p} > \frac{1-\gamma'}{1-\gamma}$, then we have $\lambda_j = \left( N^{\frac{\beta'+p}{\beta+p}} j^{-\frac{1-\gamma'}{q}} \right)^{\frac{1}{\beta'+p}}$ and thus

$$
j^{\frac{\gamma'-\gamma}{q}} \lambda_j^{\beta-\beta'} = j^{\frac{\gamma'-\gamma}{q}} \left( N^{\frac{\beta'+p}{\beta+p}} j^{-\frac{1-\gamma'}{q}} \right)^{\frac{\beta-\beta'}{\beta'+p}} = N^{\frac{\beta-\beta'}{\beta+p}} j^{\frac{1-\gamma'}{q} \left( \frac{\gamma'-\gamma}{1-\gamma'} - \frac{\beta-\beta'}{\beta'+p} \right)} \geqslant N^{\frac{\beta-\beta'}{\beta+p}}.
$$

- If $\frac{\beta'+p}{\beta+p} < \frac{1-\gamma'}{1-\gamma}$, then similarly we have $\lambda_j = \left( N^{\frac{1-\gamma'}{1-\gamma}} j^{-\frac{1-\gamma'}{q}} \right)^{\frac{1}{\beta'+p}}$ and

$$
j^{\frac{\gamma'-\gamma}{q}} \lambda_j^{\beta-\beta'} = j^{\frac{\gamma'-\gamma}{q}} \left( N^{\frac{1-\gamma'}{1-\gamma}} j^{-\frac{1-\gamma'}{q}} \right)^{\frac{\beta-\beta'}{\beta'+p}} \geqslant y_N^{\frac{\gamma'-\gamma}{q}} \left( N^{\frac{1-\gamma'}{1-\gamma}} y_N^{-\frac{1-\gamma'}{q}} \right)^{\frac{\beta-\beta'}{\beta'+p}} = N^{\frac{\gamma'-\gamma}{1-\gamma}}.
$$

Hence, in all cases (10) holds and we have that

$$
\|\mathcal{A}_0 - \mathcal{A}_\lambda\|_{\beta',\gamma'}^2 \lesssim N^{-\min\left\{ \frac{\beta-\beta'}{\beta+p}, \frac{\gamma'-\gamma}{1-\gamma} \right\}}.
\tag{11}
$$

$\square$

### B.1.2 VARIANCE

The variance term can be rewritten in the following way:

$$
\mathcal{V} = \left\| \hat{\mathcal{A}} - \mathcal{A}_\lambda \right\|_{\beta',\gamma'}^2 = \left\| C_{Q_K}^{-\frac{1-\gamma'}{2}} \left( \hat{\mathcal{A}} - \mathcal{A}_\lambda \right) C_{KK}^{\frac{1-\beta'}{2}} \right\|_{\mathrm{HS}}^2
$$

$$
= \sum_{i,j=1}^{+\infty} \left\langle \rho_j^{\frac{1}{2}} f_j, C_{Q_K}^{-\frac{1-\gamma'}{2}} \left( \hat{\mathcal{A}} - \mathcal{A}_\lambda \right) C_{KK}^{\frac{1-\beta'}{2}} \mu_i^{\frac{1}{2}} e_i \right\rangle^2 \tag{12a}
$$

$$
= \sum_{j=1}^{n_N} \rho_j^{-(1-\gamma')} \sum_{i=1}^{+\infty} \left\langle \rho_j^{\frac{1}{2}} f_j, \left[ \hat{\mathcal{C}}_{LK} \left( \hat{\mathcal{C}}_{KK} + \lambda_j I \right)^{-1} - \mathcal{C}_{LK} \left( \mathcal{C}_{KK} + \lambda_j I \right)^{-1} \right] \mu_i^{1-\frac{\beta'}{2}} e_i \right\rangle^2 \tag{12b}
$$

$$
= \sum_{j=1}^{n_N} \rho_j^{-(1-\gamma')} \sum_{i=1}^{+\infty} \Bigg\langle \underbrace{\left( \mathcal{C}_{KK} + \lambda_j I \right)^{-\frac{1}{2}} \left[ \hat{\mathcal{C}}_{KL} - \left( \hat{\mathcal{C}}_{KK} + \lambda_j I \right) \left( \mathcal{C}_{KK} + \lambda_j I \right)^{-1} \mathcal{C}_{KL} \right]}_{=:U_j} \rho_j^{\frac{1}{2}} f_j,
$$

$$
\underbrace{\left( \mathcal{C}_{KK} + \lambda_j I \right)^{\frac{1}{2}} \left( \hat{\mathcal{C}}_{KK} + \lambda_j I \right)^{-1} \left( \mathcal{C}_{KK} + \lambda_j I \right)^{\frac{1}{2}}}_{=:G_j} \frac{\mu_i^{1-\frac{\beta'}{2}}}{\sqrt{\mu_i + \lambda_j}} e_i \Bigg\rangle^2 \tag{12c}
$$

$$
= \sum_{j=1}^{n_N} \rho_j^{-(1-\gamma')} \left\langle U_j \rho_j^{\frac{1}{2}} f_j, G_j \left( \sum_{i=1}^{+\infty} \frac{\mu_i^{2-\beta'}}{\mu_i + \lambda_j} e_i \otimes e_i \right) G_j U_j \rho_j^{\frac{1}{2}} f_j \right\rangle
$$

$$
\lesssim \sum_{j=1}^{n_N} j^{\frac{1-\gamma'}{q}} \|G_j\|^2 \lambda_j^{-\beta'} \left\| U_j \rho_j^{\frac{1}{2}} f_j \right\|^2 \tag{12d}
$$

In (12), (12a) uses the definition of the Hilbert-Schmidt norm; (12b) follows from the definition of $\hat{\mathcal{A}}$ (cf.(3)) and the fact that for any $j \geqslant y_N$, we have $\left\langle \rho_j^{\frac{1}{2}} f_j, \left( \hat{\mathcal{A}} - \mathcal{A}_\lambda \right) \mu_i^{\frac{1}{2}} e_i \right\rangle = 0$; (12c) is obtained from re-arranging and (12d) follows from $\left\| \sum_{i=1}^{+\infty} \frac{\mu_i^{2-\beta'}}{\mu_i + \lambda_j} e_i \otimes e_i \right\| = \max_{i \geqslant 1} \frac{\mu_i^{1-\beta'}}{\mu_i + \lambda_j} \lesssim \lambda_j^{-\beta'}$ and $\rho_j \lesssim j^{-\frac{1}{q}}$.

Note that

$$
U_j = \left( \mathcal{C}_{KK} + \lambda_j I \right)^{-\frac{1}{2}} \left[ \hat{\mathcal{C}}_{KL} - \mathcal{C}_{KL} - \left( \hat{\mathcal{C}}_{KK} - \mathcal{C}_{KK} \right) \left( \mathcal{C}_{KK} + \lambda_j I \right)^{-1} \mathcal{C}_{KL} \right]
$$

$$
= \frac{1}{N} \sum_{k=1}^{N} \left( \mathcal{C}_{KK} + \lambda_j I \right)^{-\frac{1}{2}} \left[ u_k \otimes v_k - \mathbb{E}_{P_{KL}} u_k \otimes \mathcal{A}_0 u_k - \left( u_k \otimes u_k - \mathbb{E}_{P_{KL}} u_k \otimes u_k \right) \left( \mathcal{C}_{KK} + \lambda_j I \right)^{-1} \mathcal{C}_{KK} \mathcal{A}_0^* \right]
$$

$$
= \underbrace{\frac{1}{N} \sum_{k=1}^{N} \left( \mathcal{C}_{KK} + \lambda_j I \right)^{-\frac{1}{2}} \left( u_k \otimes \left( v_k - \mathcal{A}_0 u_k \right) \right)}_{:=U_j^1}
$$

$$
+ \underbrace{\frac{1}{N} \sum_{k=1}^{N} \left( \mathcal{C}_{KK} + \lambda_j I \right)^{-\frac{1}{2}} \left[ u_k \otimes \mathcal{A}_0 u_k - \mathbb{E}_{P_{KL}} u_k \otimes \mathcal{A}_0 u_k - \left( u_k \otimes u_k - \mathbb{E}_{P_{KL}} u_k \otimes u_k \right) \left( \mathcal{C}_{KK} + \lambda_j I \right)^{-1} \mathcal{C}_{KK} \mathcal{A}_0^* \right]}_{:=U_j^2 = \lambda_j \frac{1}{N} \sum_{k=1}^{N} \left( \mathcal{C}_{KK} + \lambda_j I \right)^{-\frac{1}{2}} \left( u_k \otimes \mathcal{A}_0 \left( \mathcal{C}_{KK} + \lambda_j I \right)^{-1} u_k - \mathbb{E}_{P_{KL}} u_k \otimes \mathcal{A}_0 \left( \mathcal{C}_{KK} + \lambda_j I \right)^{-1} u_k \right)} .
$$

The $U_j^1$ term is the variance of observational noise and $U_j^2$ term is the variance of regularized bias. Thus the $U_j^1$ term is the dominating term. Plugging the above decomposition into (12), we deduce

that $\mathcal{V} \leqslant 2\left(\mathcal{V}_1 + \mathcal{V}_2\right)$ where

$$\mathcal{V}_1 \lesssim \max_{1 \leqslant j \leqslant n_N} \|G_j\|^2 \sum_{j=1}^{n_N} j^{\frac{1-\gamma'}{q}} \lambda_j^{-\beta'} \underbrace{\left\| \frac{1}{N} \sum_{k=1}^{N} \left[ \left\langle v_k - \mathcal{A}_0 u_k, \rho_j^{\frac{1}{2}} f_j \right\rangle \left(\mathcal{C}_{KK} + \lambda_j I\right)^{-\frac{1}{2}} u_k \right] \right\|^2}_{:=\mathcal{V}_{1,j}^2}$$

$$\mathcal{V}_2 \lesssim \max_{1 \leqslant j \leqslant n_N} \|G_j\|^2 \sum_{j=1}^{n_N} j^{\frac{1-\gamma'}{q}} \lambda_j^{2-\beta'} \underbrace{\left\| \left(\hat{\mathbb{E}} - \mathbb{E}\right) \left[ \left\langle \mathcal{A}_0 \left(\mathcal{C}_{KK} + \lambda_j I\right)^{-1} u_k, \rho_j^{\frac{1}{2}} f_j \right\rangle \left(\mathcal{C}_{KK} + \lambda_j I\right)^{-\frac{1}{2}} u_k \right] \right\|^2}_{:=\mathcal{V}_{2,j}^2}$$

(13)

where $\hat{\mathbb{E}}[X] = \frac{1}{N} \sum_{k=1}^{N} X_k$ denotes the empirical mean. Define the event

$$E_{1,j} = \left\{ G_j = \left\| \left[\mathcal{P}_{i_j}\left(\mathcal{C}_{KK}\right)\right]^{\frac{1}{2}} \left[\mathcal{P}_{i_j}\left(\hat{\mathcal{C}}_{KK}\right)\right]^{\dagger} \left[\mathcal{P}_{i_j}\left(\mathcal{C}_{KK}\right)\right]^{\frac{1}{2}} \right\| \leqslant 2\sqrt{a_1}. \right\}.$$

Recall that $m_N \leqslant c_0 \left(\frac{N}{\log N}\right)^{\frac{p}{\alpha}}$, by Theorem D.3, we know that $E_{1,j}$ holds with probability $\geqslant 1 - 2e^{-a_1}$. As a result $E_1 = \cap_{j=1}^{n_N} E_{1,j}$ holds with probability $\geqslant 1 - 2n_N e^{-a_1}$. We assume event $E_1$ holds in all the following proof.

**Bounding $\mathcal{V}_1$.** Let

$$X_{j,k} = j^{\frac{1-\gamma'}{2q}} \lambda_j^{-\frac{\beta'}{2}} \left\langle v_k - \mathcal{A}_0 u_k, \rho_j^{\frac{1}{2}} f_j \right\rangle \left(\mathcal{C}_{KK} + \lambda_j I\right)^{-\frac{1}{2}} u_k \in \mathcal{H}_K$$

and $X_k = \left(X_{j,k} : 1 \leqslant j \leqslant n_N\right) \in \mathcal{H}_K^{y_N}$. Then we have $\mathcal{V}_1 \lesssim \left\| \frac{1}{N} \sum_{k=1}^{N} X_k \right\|^2$ where the norm here defined for $\mathcal{H}_K^{\otimes y_N}$ is induced by $\langle a, b \rangle = \sum_{i=1}^{n_N} \langle a_i, b_i \rangle_{\mathcal{H}_K}$. Note that $X_k, k = 1, 2, \cdots, N$ are i.i.d. random variables with mean zero, and

$$\mathbb{E} \|X_1\|^{2t} = \mathbb{E}_{P_{KL}} \left[ \left( \sum_{j=1}^{n_N} \|X_{j,k}\|^2 \right)^t \right]$$

$$= \mathbb{E}_{P_{KL}} \left[ \left( \sum_{j=1}^{n_N} j^{\frac{1-\gamma'}{q}} \lambda_j^{-\beta'} \left\langle v_1 - \mathcal{A}_0 u_1, \rho_j^{\frac{1}{2}} f_j \right\rangle^2 \left\| \left(\mathcal{C}_{KK} + \lambda_j I\right)^{-\frac{1}{2}} u \right\|^2 \right)^t \right]$$

$$\leqslant \max_{1 \leqslant j \leqslant y_N} \sup_{u \in \text{supp}(P_K)} \left( \underbrace{j^{\frac{1-\gamma'}{q}} i_j^{\frac{\beta'}{p}} \left\| \left(\mathcal{C}_{KK} + \lambda_j I\right)^{-\frac{1}{2}} u \right\|^2}_{=:G_1} \right)^{t-1} \cdot$$

$$\underbrace{\mathbb{E}_{(u,v) \sim P_{KL}} \left[ \|v - \mathcal{A}_0 u\|^{2t-2} \left( \sum_{j=1}^{n_N} j^{\frac{1-\gamma'}{q}} \lambda_j^{-\beta'} \left\langle v - \mathcal{A}_0 u, \rho_j^{\frac{1}{2}} f_j \right\rangle^2 \left\| \left(\mathcal{C}_{KK} + \lambda_j I\right)^{-\frac{1}{2}} u \right\|^2 \right) \right]}_{=:G_2}$$

By Lemma D.2 we have

$$G_1 \lesssim j^{\frac{1-\gamma'}{q}} \lambda_j^{-(\beta'+\alpha)}.$$

For $G_2$, note that for fixed $u$, Assumption 2.4 implies that

$$\mathbb{E}_{v|u} \left[ \|v - \mathcal{A}_0 u\|^{2t-2} \left( \sum_{j=1}^{n_N} j^{\frac{1-\gamma'}{q}} i_j^{\frac{\beta'}{p}} \left\langle v - \mathcal{A}_0 u, \rho_j^{\frac{1}{2}} f_j \right\rangle^2 \left\| \left(\mathcal{C}_{KK} + \lambda_j I\right)^{-\frac{1}{2}} u \right\|^2 \right) \right]$$

$$\leqslant \frac{1}{2} (2t)! R^{2t-2} \sum_{j=1}^{n_N} \sigma_j^2 j^{\frac{1-\gamma'}{q}} \lambda_j^{-\beta'} \left\| \left(\mathcal{C}_{KK} + \lambda_j I\right)^{-\frac{1}{2}} u \right\|^2.$$

where $\sigma_j^2 = \left\langle \rho_j^{\frac{1}{2}} f_j, V \rho^{\frac{1}{2}} f_j \right\rangle$. As a result, we have

$$G_2 \leqslant \mathbb{E}_{P_K} \left[ \frac{1}{2}(2t)! R^{2t-2} \sum_{j=1}^{n_N} \sigma_j^2 j^{\frac{1-\gamma'}{q}} i_j^{\frac{\beta'}{p}} \left\| (\mathcal{C}_{KK} + \lambda_j I)^{-\frac{1}{2}} u \right\|^2 \right] \leqslant \frac{1}{2}(2t)! R^{2t-2} \sigma^2 \max_{1 \leqslant j \leqslant n_N} j^{\frac{1-\gamma'}{q}} \lambda_j^{-(p+\beta')},$$

where in the second step we use $\sum_{j=1}^{+\infty} \sigma_j^2 = \operatorname{tr}(V) = \sigma^2$ and

$$\mathbb{E}_{P_K} \left[ \left\| (\mathcal{C}_{KK} + \lambda_j I)^{-\frac{1}{2}} u \right\|^2 \right] \leqslant \operatorname{tr} \left( \mathbb{E}_{P_K} \left[ (\mathcal{C}_{KK} + \lambda_j I)^{-\frac{1}{2}} u \otimes (\mathcal{C}_{KK} + \lambda_j I)^{-\frac{1}{2}} u \right] \right)$$

$$= \operatorname{tr} \left( \sum_{i=1}^{+\infty} \frac{\mu_i^2}{\mu_i + \lambda_j} e_i \otimes e_i \right)$$

$$= \sum_{i=1}^{+\infty} \frac{\mu_i}{\mu_i + \lambda_j}$$

$$\lesssim \lambda_j^{-p}.$$

We have shown that for some constant $c_1 > 0$,

$$\mathbb{E} \| X_1 \|^{2t} \leqslant \frac{1}{2}(2t)! \sigma^2 \max_{1 \leqslant j \leqslant n_N} j^{\frac{1-\gamma'}{q}} \lambda_j^{-(p+\beta')} \cdot \left( c_1 R^2 \max_{1 \leqslant j \leqslant n_N} j^{\frac{1-\gamma'}{q}} \lambda_j^{-(\beta'+p)} \right)^{t-1}.$$

By Bernstein's inequality, the event

$$E_2 := \left\{ \left\| \frac{1}{N} \sum_{k=1}^{N} X_k \right\|^2 \leqslant 6a_2 \left( \frac{\sigma^2 \max_{j \in [y_N]} j^{\frac{1-\gamma'}{q}} \lambda_j^{-(\beta'+p)}}{N} + \frac{c_1 R^2 \max_{1 \leqslant j \leqslant n_N} j^{\frac{1-\gamma'}{q}} \lambda_j^{-(\beta'+\alpha)}}{N^2} \right) \right\} \tag{14}$$

holds with probability $\geqslant 1 - 2e^{-a_2}$. By our definition of $\lambda_j$, we have

$$\max_{1 \leqslant j \leqslant n_N} j^{\frac{1-\gamma'}{q}} \lambda_j^{-(\beta'+p)} \lesssim N^{\max\left\{ \frac{\beta'+p}{\beta+p}, \frac{1-\gamma'}{1-\gamma} \right\}}$$

and $\lambda_j \gtrsim N^{-\frac{1}{\alpha}}$ (which implies that the $\frac{1}{N^2}$ term is dominated by the $\frac{1}{N}$ term). Hence, under $E_1 \cap E_2$ we have

$$\mathcal{V}_1 \lesssim a_1 a_2 \sigma^2 N^{-\min\left\{ \frac{\beta-\beta'}{\beta+p}, \frac{\gamma'-\gamma}{1-\gamma} \right\}}$$

with probability $\geqslant 1 - 2n_N e^{-a_2}$.

**Bounding $\mathcal{V}_2$.** For any $j \in \mathbb{Z}_+$ we have

$$\mathbb{E}_{u \sim P_K} \left[ \left\langle \mathcal{A}_0 (\mathcal{C}_{KK} + \lambda_j I)^{-1} u, \rho_j^{\frac{1}{2}} f_j \right\rangle^2 \right]$$

$$= \mathbb{E}_{u \sim P_K} \left\langle \rho_j^{\frac{1}{2}} f_j, \mathbb{E}_{P_K} \left[ \mathcal{A}_0 (\mathcal{C}_{KK} + \lambda_j I)^{-1} u \otimes \mathcal{A}_0 (\mathcal{C}_{KK} + \lambda_j I)^{-1} u \right] \rho_j^{\frac{1}{2}} f_j \right\rangle$$

$$= \left\langle \rho_j^{\frac{1}{2}} f_j, \mathcal{A}_0 (\mathcal{C}_{KK} + \lambda_j I)^{-1} \mathcal{C}_{KK} (\mathcal{C}_{KK} + \lambda_j I)^{-1} \mathcal{A}_0^* \rho_j^{\frac{1}{2}} f_j \right\rangle \tag{15a}$$

$$= \rho_j^{1-\gamma} \left\langle \left( \mathcal{C}_{Q_K}^{-\frac{1-\gamma}{2}} \mathcal{A}_0 \mathcal{C}_{KK}^{\frac{1-\beta}{2}} \right)^* \rho_j^{\frac{1}{2}} f_j, (\mathcal{C}_{KK} + \lambda_j I)^{-1} \mathcal{C}_{KK}^{\beta} (\mathcal{C}_{KK} + \lambda_j I)^{-1} \left( \mathcal{C}_{Q_K}^{-\frac{1-\gamma}{2}} \mathcal{A}_0 \mathcal{C}_{KK}^{\frac{1-\beta}{2}} \right)^* \rho_j^{\frac{1}{2}} f_j \right\rangle \tag{15b}$$

$$\lesssim j^{-\frac{1-\gamma}{q}} \lambda_j^{-(2-\beta)} \underbrace{\left\| \left( \mathcal{C}_{Q_K}^{-\frac{1-\gamma}{2}} \mathcal{A}_0 \mathcal{C}_{KK}^{\frac{1-\beta}{2}} \right)^* \rho_j^{\frac{1}{2}} f_j \right\|^2}_{=:D_{j,2}} \tag{15c}$$

where (15a) follows from $\mathbb{E}_{P_K} u \otimes u = \mathcal{C}_{KK}$, (15b) uses the fact that $\mathcal{C}_{KK}$ and $\mathcal{C}_{KK} + \lambda_j I$ commute, and lastly (15c) follows from $\left\| (\mathcal{C}_{KK} + \lambda_j I)^{-1} \mathcal{C}_{KK}^{\beta} (\mathcal{C}_{KK} + \lambda_j I)^{-1} \right\|_{\mathcal{H}_K} \propto \lambda_j^{-(2-\beta)}$.

Let

$$Y_{j,k} = \left\langle \mathcal{A}_0 \left( \mathcal{C}_{KK} + \lambda_j I \right)^{-1} u_k, \rho_j^{\frac{1}{2}} f_j \right\rangle \left( \mathcal{C}_{KK} + \lambda_j I \right)^{-\frac{1}{2}} u_k \in \mathcal{H}_K$$

and

$$Y_k = \left( Y_{j,k} : 1 \leqslant j \leqslant y_N \right) \in \mathcal{H}_K^{n_N}.$$

Then we have

$$\mathcal{V}_2 \lesssim \left\| \frac{1}{N} \sum_{k=1}^{N} Y_k \right\|_{\mathcal{H}_K^{y_N}}^2 .$$

Note that $Y_k, k = 1, 2, \cdots, N$ are i.i.d. random variables, and

$$\mathbb{E}\|Y_1\|^{2t} = \mathbb{E}\left[ \left( \sum_{j=1}^{n_N} \|Y_{j,k}\|^2 \right)^t \right]$$

$$= \mathbb{E}_{P_K}\left[ \left( \sum_{j=1}^{n_N} j^{\frac{1-\gamma'}{q}} \lambda_j^{2-\beta'} \left\langle \mathcal{A}_0 \left( \mathcal{C}_{KK} + \lambda_j I \right)^{-1} u_1, \rho_j^{\frac{1}{2}} f_j \right\rangle^2 \left\| \mathcal{C}_{KK}^{-\frac{1}{2}} \mathcal{I}_{i_j} \left( u_1 \right) \right\|^2 \right)^t \right]$$

$$\leqslant \sup_{u \in \mathrm{supp}(P_K)} \left( \sum_{j=1}^{n_N} j^{\frac{1-\gamma'}{q}} \lambda_j^{2-\beta'} \left\langle \mathcal{A}_0 \left( \mathcal{C}_{KK} + \lambda_j I \right)^{-1} u, \rho_j^{\frac{1}{2}} f_j \right\rangle^2 \left\| \left( \mathcal{C}_{KK} + \lambda_j I \right)^{-\frac{1}{2}} u \right\|^2 \right)^{t-1} \cdot$$

$$\sum_{j=1}^{n_N} j^{\frac{1-\gamma'}{q}} \lambda_j^{2-\beta'} \mathbb{E}\left[ \left\langle \mathcal{A}_0 \left( \mathcal{C}_{KK} + \lambda_j I \right)^{-1} u, \rho_j^{\frac{1}{2}} f_j \right\rangle^2 \right] \sup_{u \in \mathrm{supp}(P_K)} \left\| \left( \mathcal{C}_{KK} + \lambda_j I \right)^{-\frac{1}{2}} u \right\|^2$$

$$\lesssim \sup_{u \in \mathrm{supp}(P_K)} \left( \sum_{j=1}^{n_N} j^{\frac{1-\gamma'}{q}} \lambda_j^{2-\beta'-\alpha} \left\langle \mathcal{A}_0 \left( \mathcal{C}_{KK} + \lambda_j I \right)^{-1} u, \rho_j^{\frac{1}{2}} f_j \right\rangle^2 \right)^{t-1} \cdot \sum_{j=1}^{n_N} j^{\frac{1-\gamma'}{q}} \lambda_j^{-(\beta'+\alpha-\beta)} D_{j,2}.$$

$$(16)$$

For any $j \in \mathbb{Z}_+$ and $u \in \mathrm{supp}(P_K)$ we have

$$\sum_{j=1}^{n_N} j^{\frac{1-\gamma'}{q}} \lambda_j^{-(\beta'+\alpha)} \left\langle \mathcal{A}_0 \left( \mathcal{C}_{KK} + \lambda_j I \right)^{-1} \lambda_j u, \rho_j^{\frac{1}{2}} f_j \right\rangle^2$$

$$\leqslant \sum_{j=1}^{n_N} j^{\frac{1-\gamma'}{q}} \lambda_j^{-(\beta'+\alpha)} \left( 2\left\langle \mathcal{A}_0 u, \rho_j^{\frac{1}{2}} f_j \right\rangle^2 + 2\rho_j^{1-\gamma} \left\langle \mathcal{C}_{Q_K}^{-\frac{1-\gamma}{2}} \mathcal{A}_0 \left( \mathcal{C}_{KK} + \lambda_j I \right)^{-1} \mathcal{C}_{KK} u, \rho_j^{\frac{1}{2}} f_j \right\rangle^2 \right)$$

$$(17a)$$

$$\lesssim \max_{1 \leqslant j \leqslant y_N} j^{\frac{1-\gamma'}{q}} \lambda_j^{-(\beta'+\alpha)} + \sum_{j=1}^{n_N} \lambda_j^{-(\beta'+\alpha)} \lambda_j^{-\max\{\alpha-\beta,0\}} \left\| \left( \mathcal{C}_{Q_K}^{-\frac{1-\gamma}{2}} \mathcal{A}_0 \mathcal{C}_{KK}^{\frac{1-\beta}{2}} \right)^* \rho_j^{\frac{1}{2}} f_j \right\|^2 \quad (17b)$$

$$\lesssim \max_{1 \leqslant j \leqslant y_N} j^{\frac{1-\gamma'}{q}} \lambda_j^{-(\beta'+\alpha)} + \max_{1 \leqslant j \leqslant y_N} \lambda_j^{-(\beta'+\alpha)-\max\{\alpha-\beta,0\}}. \quad (17c)$$

where (17a) uses the AM-GM inequality, (17b) follows from the assumption that $\|\mathcal{A}_0 u\| \leqslant A_2$ is uniformly bounded, and that

$$\left\| \mathcal{C}_{KK}^{-\frac{1-\beta}{2}} \left( \mathcal{C}_{KK} + \lambda_j I \right)^{-1} \mathcal{C}_{KK} u \right\| = \left\| C_{KK}^{1-\frac{\alpha-\beta}{2}} \left( \mathcal{C}_{KK} + \lambda_j I \right)^{-1} \left( C_{KK}^{-\frac{1-\alpha}{2}} u \right) \right\| \lesssim \lambda_j^{-\max\{\alpha-\beta,0\}}.$$

by Assumption 2.2, and lastly (17c) follows from $\|\mathcal{A}_0\|_{\beta,\gamma} \leqslant B$.

Plugging into (16), we deduce that

$$\mathbb{E}\|Y_1\|^{2t}$$

$$\lesssim \sup_{u \in \mathrm{supp}(P_K)} \left( \max_{1 \leqslant j \leqslant n_N} j^{\frac{1-\gamma'}{q}} \lambda_j^{-(\beta'+\alpha)} + \max_{1 \leqslant j \leqslant n_N} \lambda_j^{-(\beta'+\alpha)-\max\{\alpha-\beta,0\}} \right)^{t-1} \cdot \sum_{j=1}^{n_N} j^{\frac{1-\gamma'}{q}} \lambda_j^{-(\beta'+\alpha-\beta)} D_{j,2}$$

$$\lesssim \sup_{u \in \mathrm{supp}(P_K)} \left( \max_{1 \leqslant j \leqslant n_N} j^{\frac{1-\gamma'}{q}} \lambda_j^{-(\beta'+\alpha)} + \max_{1 \leqslant j \leqslant n_N} \lambda_j^{-(\beta'+\alpha)-\max\{\alpha-\beta,0\}} \right)^{t-1} \max_{1 \leqslant j \leqslant n_N} j^{\frac{1-\gamma'}{q}} \lambda_j^{-(\beta'+\alpha-\beta)}$$

where the last step follows from $\sum_{j=1}^{+\infty} D_{j,2} = \|\mathcal{A}_0\|_{\beta,\gamma}^2$.

By Bernstein's inequality, there exists a constant $C_3$ such that the event

$$E_3 = \left\{ \mathcal{V}_2 \leqslant 6a_3 C_3 \left( \frac{j^{\frac{1-\gamma'}{q}} \lambda_j^{-(\beta'+\alpha-\beta)}}{N} + \frac{\max_{1\leqslant j\leqslant n_N} \lambda_j^{-(\beta'+\alpha)} \left( j^{\frac{1-\gamma'}{q}} + \lambda_j^{-\max\{\alpha-\beta.0\}} \right)}{N^2} \right) \right\}$$
(18)

holds with probability $\geqslant 1 - 2e^{-a_3}$.

The definition of $\lambda_N$ implies that the $\frac{1}{N^2}$ term is dominated by the $\frac{1}{N}$ term, so

$$\mathcal{V}_2 \lesssim a_1 a_3 \frac{1}{N} \max_{1\leqslant j\leqslant y_N} j^{\frac{1-\gamma}{q}} \lambda_j^{-(\beta'+\alpha-\beta)} \lesssim N^{-\min\left\{ \frac{\beta-\beta'}{\beta+p}, \frac{\gamma'-\gamma}{1-\gamma} \right\}}$$

holds under $E_1 \cap E_3$. To summarize, under $E_1 \cap E_2 \cap E_3$ which holds with probability $\geqslant 1 - 2n_N e^{-a_1} - 2e^{-a_2} - 2e^{-a_3}$, we have

$$\mathcal{V} \leqslant 2a_1 \max\{a_2, a_3\} (\mathcal{V}_1 + \mathcal{V}_2) \lesssim N^{-\min\left\{ \frac{\beta-\beta'}{\beta+p}, \frac{\gamma'-\gamma}{1-\gamma} \right\}}.$$

Recall that the bias term is upper bounded in (11). This gives the final upper bound

$$\left\| \hat{\mathcal{A}} - \mathcal{A}_0 \right\|_{\beta',\gamma'} \lesssim N^{-\min\left\{ \frac{\beta-\beta'}{2(\beta+p)}, \frac{\gamma'-\gamma}{2(1-\gamma)} \right\}}.$$

### B.1.3 THE HARD-LEARNING REGIME

In the previous sections, we focus on the case where $\alpha \leqslant \beta + p$ and establish an upper bound for the convergence rate via an optimal bias-variance trade-off. The opposite case, $\alpha > \beta + p$ is referred to as the hard-learning regime, for which the optimal rate is not known for several decades even in the case of $\gamma = 1$ (cf. the discussion following (Fischer & Steinwart, 2020, Theorem 2)). In the hard learning regime the $\mathcal{V}_2$ term becomes the leading terms.

In this section, we use the technique developed in previous sections to obtain an upper bound in the hard-learning regime. To do this, we need to re-define the truncation set $S_N$ as follows:

$$S_N = \left\{ (x,y) \in \mathbb{Z}^2 \,\middle|\, x^{\frac{\beta'+\alpha-\beta}{p}} y^{\frac{1-\gamma'}{q}} \leqslant N^{1-\min\left\{ \frac{\beta-\beta'}{\alpha}, \frac{\gamma'-\gamma}{1-\gamma} \right\}} \text{ and } x \leqslant c_0 \left( \frac{N}{\log N} \right)^{\frac{p}{\alpha}} \right\}.$$

The definition implies that the variance can be controlled by $N^{-\min\left\{ \frac{\beta-\beta'}{2\alpha}, \frac{\gamma'-\gamma}{2(1-\gamma)} \right\}}$ and it remains to focus on the bias term.

Similar to the derivations in Appendix B.1.1, we have

$$\|\mathcal{A}_0 - \mathcal{T}_N(\mathcal{A}_0)\|_{\beta',\gamma'}^2 \lesssim \max_{(i,j)\notin S_N} i^{-\frac{\beta-\beta'}{p}} j^{-\frac{\gamma'-\gamma}{q}}.$$

The maximum value of the right hand side can be achieved in either of the following two cases:

- $i = \mathcal{O}(1)$. Then we have $j \gtrsim N^{\frac{q}{1-\gamma'}\left(1-\min\left\{ \frac{\beta-\beta'}{\alpha}, \frac{\gamma'-\gamma}{1-\gamma} \right\}\right)}$ so that

$$i^{-\frac{\beta-\beta'}{p}} j^{-\frac{\gamma'-\gamma}{q}} \lesssim N^{-\frac{\gamma'-\gamma}{1-\gamma'}\left(1-\min\left\{ \frac{\beta-\beta'}{\alpha}, \frac{\gamma'-\gamma}{1-\gamma} \right\}\right)} \leqslant N^{-\frac{\gamma'-\gamma}{1-\gamma}}.$$

- $j = \mathcal{O}(1)$. In this case we must have $i \lesssim N^{\min\left\{ \frac{p}{\alpha}, \frac{p}{\beta-\beta'} \frac{\gamma'-\gamma}{1-\gamma} \right\}}$, otherwise it falls into $S_N$ by definition. Hence we have

$$i^{-\frac{\beta-\beta'}{p}} j^{-\frac{\gamma'-\gamma}{q}} \leqslant i^{-\frac{\beta-\beta'}{p}} \lesssim N^{-\min\left\{ \frac{\beta-\beta'}{\alpha}, \frac{\gamma'-\gamma}{1-\gamma} \right\}}.$$

On the other hand, for the variance term we still have $\mathcal{V}_1 \lesssim \frac{1}{N} \max_{1\leqslant j\leqslant n_N} j^{\frac{1-\gamma'}{q}} i_j^{\frac{\beta'+p}{p}}$ and $\mathcal{V}_2 \leqslant \frac{1}{N} \max_{1\leqslant j\leqslant n_N} j^{\frac{1-\gamma'}{q}} i_j^{\frac{\beta'+\alpha-\beta}{p}}$, so that

$$\mathcal{V} \lesssim \frac{1}{N} \max_{1\leqslant j\leqslant n_N} j^{\frac{1-\gamma'}{q}} i_j^{\frac{\beta'+\alpha-\beta}{p}} \leqslant N^{-\min\left\{ \frac{\beta-\beta'}{\beta+p}, \frac{\gamma'-\gamma}{1-\gamma} \right\}}.$$

As a result, we can obtain the following convergence rate:

$$\left\| \hat{\mathcal{A}} - \mathcal{A}_0 \right\|_{\beta',\gamma'} \lesssim N^{-\min\left\{ \frac{\beta-\beta'}{2\alpha}, \frac{\gamma'-\gamma}{2(1-\gamma)} \right\}}.$$

## B.2 REGULARIZATION VIA BIAS CONTOUR

In this subsection, we analyze the convergence rate of regularization via bias contour (cf. Figure 2). Specifically, we consider the estimator (3) with the choice

$$\lambda_j = \max\left\{ \left( j^{-\frac{\gamma'-\gamma}{q}} N^{\min\left\{ \frac{\beta-\beta'}{\max\{\alpha,\beta+p\}}, \frac{\gamma'-\gamma}{1-\gamma} \right\}} \right)^{-\frac{1}{\beta-\beta'}}, c_0 \left( \frac{N}{\log N} \right)^{-\frac{1}{\alpha}} \right\}. \tag{19}$$

It now remains to plug the above $\lambda_j$ into our bounds for bias and variance derived in the previous subsections.

**Bounding the bias term.** It follows from (9) that

$$\|\mathcal{A}_0 - \mathcal{A}_\lambda\|_{\beta,\gamma'}^2 \lesssim \max_{1 \leqslant j \leqslant y_N} j^{-\frac{\gamma'-\gamma}{q}} \lambda_j^{\beta-\beta'}$$

$$\lesssim \max\left\{ N^{-\min\left\{ \frac{\beta-\beta'}{\max\{\alpha,\beta+p\}}, \frac{\gamma'-\gamma}{1-\gamma} \right\}}, c_0 \left( \frac{N}{\log N} \right)^{-\frac{\beta-\beta'}{\alpha}} \right\}$$

$$\lesssim N^{-\min\left\{ \frac{\beta-\beta'}{\max\{\alpha,\beta+p\}}, \frac{\gamma'-\gamma}{1-\gamma} \right\}}.$$

**Bounding the variance term** It follows from (14) and (18) that the variance is bounded by

$$\left\| \hat{\mathcal{A}} - \mathcal{A}_\lambda \right\|_{\beta',\gamma'}^2 \lesssim \frac{1}{N} \max_{1 \leqslant j \leqslant y_N} j^{\frac{1-\gamma'}{q}} \lambda_j^{-\left(\beta'+\max\{\alpha-\beta,p\}\right)}.$$

As before, we consider the cases $\alpha \leqslant \beta + p$ and $\alpha > \beta + p$ separately.

- If $\alpha \leqslant \beta + p$, then it follows that

$$\left\| \hat{\mathcal{A}} - \mathcal{A}_\lambda \right\|_{\beta',\gamma'}^2 \lesssim \frac{1}{N} \max_{1 \leqslant j \leqslant y_N} j^{\frac{1-\gamma'}{q}} \lambda_j^{-(\beta'+p)}$$

$$\lesssim \frac{1}{N} \max_{1 \leqslant j \leqslant y_N} j^{\frac{1-\gamma'}{q}} \left( j^{-\frac{\gamma'-\gamma}{q}} N^{\min\left\{ \frac{\beta-\beta'}{\beta+p}, \frac{\gamma'-\gamma}{1-\gamma} \right\}} \right)^{\frac{\beta'+p}{\beta-\beta'}}$$

$$\lesssim \frac{1}{N} \max_{1 \leqslant j \leqslant y_N} j^{\frac{\gamma'-\gamma}{q} \left( \frac{1-\gamma'}{\gamma'-\gamma} - \frac{\beta'+p}{\beta-\beta'} \right)} N^{\frac{\beta'+p}{\beta-\beta'} \min\left\{ \frac{\beta-\beta'}{\beta+p}, \frac{\gamma'-\gamma}{1-\gamma} \right\}}$$

$$= \frac{1}{N} \max_{j \in \{1, y_N\}} j^{\frac{\gamma'-\gamma}{q} \left( \frac{1-\gamma'}{\gamma'-\gamma} - \frac{\beta'+p}{\beta-\beta'} \right)} N^{\frac{\beta'+p}{\beta-\beta'} \min\left\{ \frac{\beta-\beta'}{\beta+p}, \frac{\gamma'-\gamma}{1-\gamma} \right\}}$$

$$= N^{\min\left\{ \frac{\beta-\beta'}{\beta+p}, \frac{\gamma'-\gamma}{1-\gamma} \right\} \max\left\{ \frac{\beta'+p}{\beta-\beta'}, \frac{1-\gamma'}{\gamma'-\gamma} \right\} - 1}$$

$$= N^{-\min\left\{ \frac{\beta-\beta'}{\beta+p}, \frac{\gamma'-\gamma}{1-\gamma} \right\}},$$

  where we use $y_N^{\frac{\gamma'-\gamma}{q}} = N^{\min\left\{ \frac{\beta-\beta'}{\beta+p}, \frac{\gamma'-\gamma}{1-\gamma} \right\}}$ by definition.

- If $\alpha > \beta + p$, then similarly we have

$$\left\| \hat{\mathcal{A}} - \mathcal{A}_\lambda \right\|_{\beta',\gamma'}^2 \lesssim \frac{1}{N} \max_{1 \leqslant j \leqslant y_N} j^{\frac{1-\gamma'}{q}} \lambda_j^{-\beta'+\alpha-\beta}$$

$$\lesssim \frac{1}{N} \max_{1 \leqslant j \leqslant y_N} j^{\frac{1-\gamma'}{q}} \left( j^{-\frac{\gamma'-\gamma}{q}} N^{\min\left\{ \frac{\beta-\beta'}{\alpha}, \frac{\gamma'-\gamma}{1-\gamma} \right\}} \right)^{\frac{\beta'+\alpha-\beta}{\beta-\beta'}}$$

$$= \frac{1}{N} \max_{j \in \{1, y_N\}} j^{\frac{1-\gamma'}{q}} \left( j^{-\frac{\gamma'-\gamma}{q}} N^{\min\left\{ \frac{\beta-\beta'}{\alpha}, \frac{\gamma'-\gamma}{1-\gamma} \right\}} \right)^{\frac{\beta'+\alpha-\beta}{\beta-\beta'}}$$

$$\leqslant N^{-\min\left\{ \frac{\beta-\beta'}{\alpha}, \frac{\gamma'-\gamma}{1-\gamma} \right\}}.$$

Hence we deduce that

$$\left\| \hat{\mathcal{A}} - \mathcal{A}_\lambda \right\|_{\beta',\gamma'}^2 \lesssim N^{-\min\left\{ \frac{\beta-\beta'}{\alpha}, \frac{\gamma'-\gamma}{1-\gamma} \right\}},$$

as desired.

### B.3 IMPLICATION OF THE UPPER BOUND

In this section, we discuss the implications of our upper bounds under the $(\beta', \gamma')$-norm.

Note that $\left\| \mathcal{C}_{Q_K}^{-\frac{1-\gamma'}{2}} v \right\|_{\mathcal{H}_L} = \|v\|_{\mathcal{H}_L^{2-\gamma'}}$ for all $v \in L_2(Q_K)$ (if one side of the equation is $+\infty$ then so is the other), we have that

$$
\begin{aligned}
\mathbb{E}_{u \sim P_K} \left\| \left( \hat{\mathcal{A}} - \mathcal{A}_0 \right) u \right\|_{\mathcal{H}_L^{2-\gamma'}}^2 &= \mathbb{E}_{u \sim P_K} \left\| \mathcal{C}_{Q_K}^{-\frac{1-\gamma'}{2}} \left( \hat{\mathcal{A}} - \mathcal{A}_0 \right) u \right\|_{\mathcal{H}_L}^2 \\
&= \mathrm{tr} \left( \mathcal{C}_{Q_K}^{-\frac{1-\gamma'}{2}} \left( \hat{\mathcal{A}} - \mathcal{A}_0 \right) \mathbb{E}_{u \sim P_K} u \otimes u \left( \mathcal{C}_{Q_K}^{-\frac{1-\gamma'}{2}} \left( \hat{\mathcal{A}} - \mathcal{A}_0 \right) \right)^* \right) \\
&\lesssim \left\| \hat{\mathcal{A}} - \mathcal{A}_0 \right\|_{\beta',\gamma'}^2,
\end{aligned}
$$

(20)

where the last step follows from $\mathbb{E}_{u \sim P_K} u \otimes u = \mathcal{C}_{P_K}$. Note that the above derivations hold for any $0 \leqslant \beta' < \beta$, so choosing $\beta' = 0$ yields the best upper bound. We can see from (20) that our analysis implies an upper bound of the expected error of the learned solution evaluated under the $\mathcal{H}_L^{2-\gamma'}$ norm. On the other hand, it is also possible to obtain a *uniform* convergence rate when $\beta' \geqslant \alpha$:

$$
\begin{aligned}
\left\| \left( \hat{\mathcal{A}} - \mathcal{A}_0 \right) u \right\|_{\mathcal{H}_L^{2-\gamma'}} &= \left\| \mathcal{C}_{Q_K}^{-\frac{1-\gamma'}{2}} \left( \hat{\mathcal{A}} - \mathcal{A}_0 \right) u \right\|_{\mathcal{H}_L} \\
&\leqslant \left\| \hat{\mathcal{A}} - \mathcal{A}_0 \right\|_{\beta',\gamma'} \cdot \left\| \mathcal{C}_{P_K}^{-\frac{1-\beta'}{2}} u \right\|_{\mathcal{H}_K} \lesssim \left\| \hat{\mathcal{A}} - \mathcal{A}_0 \right\|_{\beta',\gamma'}.
\end{aligned}
$$

## C PROOFS FOR THE MULTI-LEVEL OPERATOR LEARNING ALGORITHM

In this section, we analyze the convergence rate of our multi-level algorithm described in Section 5. We define $\eta_1 = \min \left\{ \frac{\beta-\beta'}{\max\{\alpha,\beta+p\}}, \frac{\gamma'-\gamma}{1-\gamma} \right\}$ and $\eta_2 = \max \left\{ 1 - \frac{\beta-\beta'}{\max\{\alpha,\beta+p\}}, \frac{1-\gamma'}{1-\gamma} \right\} = 1 - \eta_1$. We first restrict ourselves to the case when $\frac{\beta-\beta'}{\max\{\alpha,\beta+p\}} \neq \frac{\gamma'-\gamma}{1-\gamma}$; the special case when the two terms are equal will be separately treated in Appendix C.1. For the optimal bias and variance contours $\ell_{C_1,\text{bias}}$ and $\ell_{C_2,\text{var}}$ with $C_1 = N^{\eta_1}$ and $C_2 = N^{\eta_2}$, we define a sequence $\{x_n\}$ as follows:

$$x_0 = \max \left\{ \frac{1}{2} N^{\frac{p}{\beta'+p} \eta_2}, c_0 \left( \frac{N}{\log N} \right)^{-\frac{1}{\alpha}} \right\} \tag{21a}$$

$$y_n = \text{the solution of } x_n^{\frac{\beta'+\max\{\alpha-\beta,p\}}{p}} y^{\frac{1-\gamma'}{q}} = N^{\eta_2}, \quad n \geqslant 0 \tag{21b}$$

$$x_{n+1} = \text{the solution of } x^{\frac{\beta-\beta'}{p}} y_n^{\frac{\gamma'-\gamma}{q}} = N^{\eta_1}, \quad n \geqslant 0. \tag{21c}$$

We first derive an explicit recursive formula for $\{x_n\}$.

**Lemma C.1** *Let* $u = \frac{\beta'+\max\{\alpha-\beta,p\}}{\beta-\beta'} \frac{\gamma'-\gamma}{1-\gamma'} > 0$, *then*

*(1). if $u > 1$, then*

$$N^{-\frac{p}{\beta+p}} x_{n+1} = \left( N^{-\frac{p}{\beta+p}} x_n \right)^u.$$

*(2). if $u < 1$, then*

$$x_{n+1} = x_n^u.$$

***Proof***:

(1). Suppose that $u > 1$, then we have $\eta_1 = \frac{\beta - \beta'}{\max\{\alpha, \beta + p\}}$ and $\eta_2 = 1 - \eta_1$. It follows from (21b) and (21c) that

$$
\begin{aligned}
x_{n+1} &= N^{\frac{p}{\max\{\alpha, \beta + p\}}} y_n^{-\frac{\gamma' - \gamma}{q} \frac{p}{\beta - \beta'}} \\
&= N^{\frac{p}{\max\{\alpha, \beta + p\}}} \left( N^{\eta_2} x_n^{-\frac{\beta' + \max\{\alpha - \beta, p\}}{p}} \right)^{-\frac{\gamma' - \gamma}{1 - \gamma'} \frac{p}{\beta - \beta'}} \\
&= N^{\frac{p}{\max\{\alpha, \beta + p\}}} \left( N^{-\frac{p}{\max\{\alpha, \beta + p\}}} x_n \right)^u.
\end{aligned}
$$

(2). Suppose that $u < 1$, then we have $\eta_1 = \frac{\gamma' - \gamma}{1 - \gamma}$ and $\eta_2 = \frac{1 - \gamma'}{1 - \gamma}$, so that $\frac{\eta_1}{\eta_2} = \frac{\gamma' - \gamma}{1 - \gamma'}$, and it follows from (21b) and (21c) that $x_n^{\frac{\beta' + \max\{\alpha - \beta, p\}}{p} \frac{\gamma' - \gamma}{1 - \gamma'}} = x_{n+1}^{\frac{\beta - \beta'}{p}}$, thus $x_{n+1} = x_n^u$.

$\square$

Lemma C.1 implies that when $u \neq 1$, the sequence $\{x_n\}$ decreases super-exponentially. Thus, there exists $L_N = \mathcal{O}(\log \log N)$ such that $x_n \leqslant 2$ for all $n \geqslant L_N$.

Let $\lambda_i^{(K)} = x_i^{-\frac{1}{p}}$ and $\lambda_i^{(L)} = y_i^{-\frac{1}{q}}$, then we construct the following estimator:

$$
\hat{\mathcal{A}}_{\mathtt{ml}} = \sum_{i=0}^{L_N} \left( \sum_{y_{i-1} \leqslant j < y_i} \rho_j^{\frac{1}{2}} f_j \otimes \rho_j^{\frac{1}{2}} f_j \right) \hat{\mathcal{C}}_{YX} \left( \hat{\mathcal{C}}_{KK} + \lambda_i^{(K)} I \right)^{-1} \tag{22}
$$

where $y_{-1} := 0$. Note that each summand in the above equation is essentially a regularized least-squares estimator and learns a rectangular region. The following theorem states that the estimator $\hat{\mathcal{A}}_{\mathtt{ml}}$ can achieve minimax optimal convergence rate.

**Theorem C.1** *Consider the estimator $\hat{\mathcal{A}}_{\mathtt{ml}}$ defined by (5). Suppose that Assumptions 2.1 to 2.5 hold, then there exists a universal constant $C$, such that*

$$
\left\| \hat{\mathcal{A}}_{\mathtt{ml}} - \mathcal{A}_0 \right\|_{\beta', \gamma'}^2 \leqslant C \tau^2 \left( \frac{N}{\log N} \right)^{-\min\left\{ \frac{\beta - \beta'}{\max\{\alpha, \beta + p\}}, \frac{\gamma' - \gamma}{1 - \gamma} \right\}} \log^2 N
$$

*holds with probability $\geqslant 1 - e^{-\tau}$.*

***Proof***: The proof of Theorem 5.1 is similar to that of Theorems 4.1 and 4.2. We consider the bias-variance decomposition of the estimation error

$$
\left\| \hat{\mathcal{A}}_{\mathtt{ml}} - \mathcal{A}_0 \right\|_{\beta', \gamma'} \leqslant \left\| \hat{\mathcal{A}}_{\mathtt{ml}} - \hat{\mathcal{A}}_{\mathtt{ml}}^{\lambda} \right\|_{\beta', \gamma'} + \left\| \hat{\mathcal{A}}_{\mathtt{ml}}^{\lambda} - \mathcal{A}_0 \right\|_{\beta', \gamma'}
$$

where

$$
\hat{\mathcal{A}}_{\mathtt{ml}}^{\lambda} = \sum_{i=0}^{L_N} \left( \sum_{y_i \leqslant j < y_{i+1}} \rho_j^{\frac{1}{2}} f_j \otimes \rho_j^{\frac{1}{2}} f_j \right) \mathcal{C}_{YX} \left( \mathcal{C}_{KK} + \lambda_i^{(K)} I \right)^{-1}. \tag{23}
$$

**Bounding the bias term.** Since $\|\mathcal{A}_0\|_{\beta, \gamma} \leqslant B$, we can write

$$
\mathcal{A}_0 := \sum_{i=1}^{+\infty} \sum_{j=1}^{+\infty} a_{ij} \mu_i^{\frac{\beta}{2}} \rho_j^{1 - \frac{\gamma}{2}} f_j \otimes e_i
$$

where the coefficient matrix $A_0 = (a_{ij})_{1 \leqslant i, j \leqslant +\infty}$ satisfies $\|A_0\|_F^2 \leqslant B^2$. We fix $(i, j) \in \mathbb{Z}_+^2$ and assume WLOG that $y_{m_j - 1} \leqslant j < y_{m_j}$ for some $m \geqslant 0$, where $y_{L_N + 1} = +\infty$. It follows from (23) that

$$
\begin{aligned}
\left\langle \rho_j^{\frac{1}{2}} f_j, \hat{\mathcal{A}}_{\mathtt{ml}}^{\lambda} \mu_i^{\frac{1}{2}} e_i \right\rangle &= \sum_{k=0}^{L_N} \left\langle \left( \sum_{y_{k-1} \leqslant j < y_k} \rho_j^{\frac{1}{2}} f_j \otimes \rho_j^{\frac{1}{2}} f_j \right) \rho_j^{\frac{1}{2}} f_j, \mathcal{C}_{YX} \left( \mathcal{C}_{KK} + \lambda_k^{(K)} I \right)^{-1} \mu_i^{\frac{1}{2}} e_i \right\rangle \\
&= \frac{\mu_i}{\mu_i + \lambda_m^{(K)}} \rho_j^{\frac{1 - \gamma}{2}} \mu_i^{-\frac{1 - \beta}{2}} a_{ij}.
\end{aligned}
$$

Thus

$$
\begin{aligned}
\left\| \mathcal{A}_0 - \hat{\mathcal{A}}_{\mathtt{ml}}^\lambda \right\|_{\beta',\gamma'}^2 &= \left\| \mathcal{C}_{Q_L}^{-\frac{1-\gamma}{2}} \left( \hat{\mathcal{A}}_{\mathtt{ml}}^\lambda - \mathcal{A}_0 \right) \mathcal{C}_{K\tilde{K}}^{\frac{1-\beta'}{2}} \right\|_{\mathrm{HS}}^2 \\
&= \sum_{i,j=1}^{+\infty} \left\langle \rho_j^{\frac{1}{2}} f_j, \mathcal{C}_{Q_L}^{-\frac{1-\gamma'}{2}} \left( \hat{\mathcal{A}}_{\mathtt{ml}}^\lambda - \mathcal{A}_0 \right) \mathcal{C}_{K\tilde{K}}^{\frac{1-\beta'}{2}} \mu_i^{\frac{1}{2}} e_i \right\rangle^2 \\
&= \sum_{i,j=1}^{+\infty} \left( \frac{\lambda_{m_j}^{(K)}}{\mu_i + \lambda_{m_j}^{(K)}} \right)^2 \mu_i^{\beta-\beta'} \rho_j^{\gamma'-\gamma} a_{ij}^2 \\
&= \sum_{j=1}^{+\infty} \rho_j^{\gamma'-\gamma} \left( \sum_{i=1}^{+\infty} a_{ij}^2 \right) \max_{i\geqslant 1} \mu_i^{\beta-\beta'} \left( \frac{\lambda_{m_j}^{(K)}}{\mu_i + \lambda_{m_j}^{(K)}} \right)^2 \qquad (24) \\
&\lesssim \sum_{j=1}^{+\infty} \rho_j^{\gamma'-\gamma} \left( \lambda_{m_j}^{(K)} \right)^{\beta-\beta'} \left( \sum_{i=1}^{+\infty} a_{ij}^2 \right) \lesssim B^2 \max_{j\geqslant 1} \rho_j^{\gamma'-\gamma} \left( \lambda_{m_j}^{(K)} \right)^{\beta-\beta'} \\
&\leqslant B^2 \max_{j\geqslant 1} \rho_j^{\gamma'-\gamma} x_{m_j}^{-\frac{\beta-\beta'}{p}} \lesssim B^2 \max_{j\geqslant 1} j^{-\frac{\gamma'-\gamma}{q}} x_{m_j}^{-\frac{\beta-\beta'}{p}} \\
&\leqslant B^2 y_{m_j-1}^{-\frac{\gamma'-\gamma}{q}} x_{m_j}^{-\frac{\beta-\beta'}{p}} \lesssim N^{-\eta_1}
\end{aligned}
$$

where we recall that $\eta_1 = \min\left\{ \frac{\beta-\beta'}{\max\{\alpha,\beta+p\}}, \frac{\gamma'-\gamma}{1-\gamma} \right\}$ and the last step follows from (21c).

**Bounding the variance term.** The variance term can be rewritten in the following way:

$$
\begin{aligned}
\mathcal{V} &= \left\| \hat{\mathcal{A}}_{\mathtt{ml}} - \mathcal{A}_{\mathtt{ml}}^\lambda \right\|_{\beta',\gamma'}^2 \\
&= \left\| C_{Q_L}^{-\frac{1-\gamma'}{2}} \left( \hat{\mathcal{A}}_{\mathtt{ml}} - \mathcal{A}_{\mathtt{ml}}^\lambda \right) C_{K\tilde{K}}^{\frac{1-\beta'}{2}} \right\|_{\mathrm{HS}}^2 \\
&= \sum_{i,j=1}^{+\infty} \left\langle \rho_j^{\frac{1}{2}} f_j, C_{Q_L}^{-\frac{1-\gamma'}{2}} \left( \hat{\mathcal{A}}_{\mathtt{ml}} - \mathcal{A}_{\mathtt{ml}}^\lambda \right) C_{K\tilde{K}}^{\frac{1-\beta'}{2}} \mu_i^{\frac{1}{2}} e_i \right\rangle^2 \\
&= \sum_{j=1}^{z_N} \rho_j^{-(1-\gamma')} \sum_{i=1}^{+\infty} \left\langle \rho_j^{\frac{1}{2}} f_j, \left[ \hat{\mathcal{C}}_{YX} \left( \hat{\mathcal{C}}_{KK} + \lambda_{m_j} I \right)^{-1} - \mathcal{C}_{YX} \left( \mathcal{C}_{KK} + \lambda_{m_j} I \right)^{-1} \right] \mu_i^{1-\frac{\beta'}{2}} e_i \right\rangle^2 \\
&= \sum_{j=1}^{z_N} \rho_j^{-(1-\gamma')} \sum_{i=1}^{+\infty} \left\langle \underbrace{\left( \mathcal{C}_{KK} + \lambda_{m_j} I \right)^{-\frac{1}{2}} \left[ \hat{\mathcal{C}}_{KL} - \left( \hat{\mathcal{C}}_{KK} + \lambda_{m_j} I \right) \left( \mathcal{C}_{KK} + \lambda_{m_j} I \right)^{-1} \mathcal{C}_{KL} \right]}_{=:U_{m_j}} \rho_j^{\frac{1}{2}} f_j, \right.
\end{aligned}
$$

$$
\left. \underbrace{\left( \mathcal{C}_{KK} + \lambda_{m_j} I \right)^{\frac{1}{2}} \left( \hat{\mathcal{C}}_{KK} + \lambda_{m_j} I \right)^{-1} \left( \mathcal{C}_{KK} + \lambda_{m_j} I \right)^{\frac{1}{2}}}_{=:G_{m_j}} \frac{\mu_i^{1-\frac{\beta'}{2}}}{\sqrt{\mu_i + \lambda_j}} e_i \right\rangle^2
$$

$$
\begin{aligned}
&= \sum_{j=1}^{z_N} \rho_j^{-(1-\gamma')} \left\langle U_{m_j} \rho_j^{\frac{1}{2}} f_j, G_{m_j} \left( \sum_{i=1}^{+\infty} \frac{\mu_i^{2-\beta'}}{\mu_i + \lambda_{m_j}} e_i \otimes e_i \right) G_{m_j} U_{m_j} \rho_j^{\frac{1}{2}} f_j \right\rangle \\
&\lesssim \sum_{j=1}^{z_N} j^{\frac{1-\gamma'}{q}} \| G_{m_j} \|^2 \lambda_{m_j}^{-\beta'} \left\| U_{m_j} \rho_j^{\frac{1}{2}} f_j \right\|^2
\end{aligned}
$$

for reasons similar to (12). It now remains to bound $\left\| G_{m_j} \right\|$ and $\left\| U_{m_j} \rho_j^{\frac{1}{2}} f_j \right\|$ for $1 \leqslant j \leqslant L_N$. Note that these quantities have already been bounded in Appendix B.1.2 with $\lambda_{m_j}$ replaced with $\lambda_j$ (there we use a different regularization for each $j$). Hence, those bounds can be directly applied here, so there exists a constant $C > 0$ such that

$$
\mathcal{V} \leqslant C a^2 \frac{1}{N} \max_{1\leqslant j\leqslant L_N} j^{\frac{1-\gamma'}{q}} \lambda_{m_j}^{-\left(\beta' + \max\{\alpha-\beta, p\}\right)}
$$

with probability $\geqslant 1 - Ne^{-a}$. Since $j \leqslant y_{m_j}$, by (21b) we have

$$j^{\frac{1-\gamma'}{q}} \lambda_{m_j}^{-\left(\beta' + \max\{\alpha-\beta, p\}\right)} \lesssim y_{m_j}^{\frac{1-\gamma'}{q}} x_{m_j}^{\frac{\beta' + \max\{\alpha-\beta, p\}}{p}} = N^{\eta_2}.$$

Hence

$$\mathcal{V} \lesssim \frac{1}{N} \max_{1 \leqslant j \leqslant L_N} j^{\frac{1-\gamma'}{q}} \lambda_{m_j}^{-\left(\beta' + \max\{\alpha-\beta, p\}\right)} \leqslant N^{\eta_2 - 1} = N^{\eta_1}.$$

Combining the bias and variance bounds, the conclusion directly follows. $\qquad\square$

## C.1   Special case: $\frac{\beta - \beta'}{\max\{\alpha, \beta+p\}} = \frac{\gamma' - \gamma}{1 - \gamma}$

Note that Lemma C.1 does not cover the case $u = 1$, or equivalently $\frac{\beta-\beta'}{\max\{\alpha,\beta+p\}} = \frac{\gamma'-\gamma}{1-\gamma}$. This case is special since the bias contour coincides with the variance contour, and we need to modify our construction of the multilevel estimator.

We define two sequences $\{x_n\}, \{y_n\}$ as follows:

$$x_0 = \max\left\{ \frac{1}{2} N^{\frac{p}{\beta'+p}\eta_2}, c_0 \left( \frac{N}{\log N} \right)^{-\frac{1}{\alpha}} \right\}$$

$$x_n = \frac{1}{2} x_{n-1} \tag{25}$$

$$y_n = \text{the solution of } x_n^{\frac{\beta-\beta'}{p}} y_n^{\frac{\gamma'-\gamma}{q}} = N^{\eta_1},$$

where we recall that $\eta_1 = \frac{\beta-\beta'}{\max\{\alpha,\beta+p\}} = \frac{\gamma'-\gamma}{1-\gamma}$. In this case, there exists $L_N = \mathcal{O}(\ln N)$ such that $x_n < 1$ for all $n \geqslant L_N$. Let $\lambda_i^{(K)} = x_i^{-\frac{1}{p}}$, then we construct the following estimator:

$$\hat{\mathcal{A}}_{\texttt{ml}}^{\lambda} = \sum_{i=0}^{L_N} \left( \sum_{y_{i-1} \leqslant j < y_i} \rho_j^{\frac{1}{2}} f_j \otimes \rho_j^{\frac{1}{2}} f_j \right) \hat{\mathcal{C}}_{LK} \left( \hat{\mathcal{C}}_{KK} + \lambda_i^{(K)} I \right)^{-1}. \tag{26}$$

Similar to Theorem C.1, we can establish the following result:

**Theorem C.2** *Consider the estimator $\hat{\mathcal{A}}_{\texttt{ml}}$ defined by (26). Suppose that Assumptions 2.1 to 2.5 hold, then there exists a universal constant $C$, such that*

$$\left\| \hat{\mathcal{A}}_{\texttt{ml}} - \mathcal{A}_0 \right\|_{\beta',\gamma'}^2 \leqslant C\tau^2 \left( \frac{N}{\log N} \right)^{-\min\left\{ \frac{\beta-\beta'}{\max\{\alpha,\beta+p\}}, \frac{\gamma'-\gamma}{1-\gamma} \right\}} \log^2 N$$

*holds with probability $\geqslant 1 - e^{-\tau}$.*

***Proof:*** The proof of Theorem C.2 is similar to that of Theorems 4.1 and 4.2. We consider the bias-variance decomposition

$$\left\| \hat{\mathcal{A}}_{\texttt{ml}} - \mathcal{A}_0 \right\|_{\beta',\gamma'} \leqslant \left\| \hat{\mathcal{A}}_{m1} - \hat{\mathcal{A}}_{m1}^{\lambda} \right\|_{\beta',\gamma'} + \left\| \hat{\mathcal{A}}_{m1}^{\lambda} - \mathcal{A}_0 \right\|_{\beta',\gamma'}$$

where

$$\mathcal{A}_{\texttt{ml}}^{\lambda} = \sum_{i=0}^{L_N} \left( \sum_{y_{i-1} \leqslant j < y_i} \rho_j^{\frac{1}{2}} f_j \otimes \rho_j^{\frac{1}{2}} f_j \right) \mathcal{C}_{LK} \left( \mathcal{C}_{KK} + \lambda_i^{(K)} I \right)^{-1}. \tag{27}$$

as defined in (26).

**Bounding the bias term.** Let $\mathcal{A}_0 := \sum_{i=1}^{+\infty} \sum_{j=1}^{+\infty} a_{ij} \mu_i^{\frac{\beta}{2}} \rho_j^{\frac{1-\gamma}{2}} f_j \otimes e_i$ with coefficient matrix $A_0 = (a_{ij})_{i,j=1}^{+\infty}$ such that $\|A_0\|_F^2 \leqslant B^2$. We fix $(i,j) \in \mathbb{Z}_+^2$ and assume WLOG that $y_{m_j-1} \leqslant j < y_{m_j}$ for some $m_j \geqslant 0$, where $y_{L_N+1} = +\infty$. It follows from (27) that

$$\left\langle \rho_j^{\frac{1}{2}} f_j, \mathcal{A}_{\texttt{ml}}^{\lambda} \mu_i^{\frac{1}{2}} e_i \right\rangle = \frac{\mu_i}{\mu_i + \lambda_{m_j}^{(K)}} \rho_j^{\frac{1-\gamma}{2}} \mu_i^{-\frac{1-\beta}{2}} a_{ij}.$$

Thus we can proceed as in (24) to deduce that

$$\left\| \mathcal{A}_0 - \mathcal{A}_{\mathrm{m1}}^\lambda \right\|_{\beta',\gamma'}^2 \leqslant \max_{j\geqslant 1} \rho_j^{\gamma'-\gamma} \left( \lambda_{m_j}^{(K)} \right)^{\beta-\beta'}$$

$$\lesssim \max \left\{ \max_{1\leqslant j\leqslant L_N} j^{-\frac{\gamma'-\gamma}{q}} x_{m_j}^{-\frac{\beta-\beta'}{p}}, y_{L_N}^{-\frac{\gamma'-\gamma}{q}} \right\}$$

$$\leqslant \max \left\{ \max_{1\leqslant j\leqslant L_N} y_{m_j-1}^{-\frac{\gamma'-\gamma}{q}} x_{m_j}^{-\frac{\beta-\beta'}{p}}, y_{L_N}^{-\frac{\gamma'-\gamma}{q}} \right\}$$

The definition (25) implies that

$$y_{m_j-1}^{-\frac{\gamma'-\gamma}{q}} x_{m_j}^{-\frac{\beta-\beta'}{p}} \leqslant 2^{\frac{\beta-\beta'}{p}} y_{m_j-1}^{-\frac{\gamma'-\gamma}{q}} x_{m_j-1}^{-\frac{\beta-\beta'}{p}} \leqslant 2^{\frac{\beta-\beta'}{p}} N^{-\eta_1}.$$

On the other hand, since $x_{L_N} < 1$, by (25) implies that $y_{L_N}^{-\frac{\gamma'-\gamma}{q}} \lesssim N^{-\eta_1}$. Therefore, for the bias term $\left\| \mathcal{A}_0 - \mathcal{A}_{\mathrm{m1}}^\lambda \right\|_{\beta',\gamma'}^2 \lesssim N^{-\eta_1}$.

**Bounding the variance term.** Repeating the arguments in (25), we can deduce that there exists a constant $C > 0$ such that

$$\mathcal{V} \leqslant Ca^2 \frac{1}{N} \max_{1\leqslant j\leqslant L_N} j^{\frac{1-\gamma'}{q}} \lambda_{m_j}^{-\left(\beta'+\max\{\alpha-\beta,p\}\right)} \leqslant Ca^2 \frac{1}{N} \max_{1\leqslant j\leqslant L_N} y_{m_j}^{\frac{1-\gamma'}{q}} x_{m_j}^{\frac{(\beta'+\max\{\alpha-\beta,p\})}{p}} \lesssim N^{-\eta_1}$$

with probability $\geqslant 1 - Ne^{-a}$.

Combining the bias and variance bounds, we arrive at the desired conclusion. □

The conclusion of Theorem 5.1 then follows from Theorems C.1 and C.2.

## D    AUXILIARY RESULTS

**Lemma D.1** *We have* $\|T\|_{\beta,\gamma} = \left\| \mathcal{C}_{Q_L}^{-(1-\gamma)/2} \circ T \circ \mathcal{C}_{KK}^{(1-\beta)/2} \right\|_{\mathrm{HS}(\mathcal{H}_K,\mathcal{H}_L)}.$

**Proof**: We recall from the definition that $\|T\|_{\beta,\gamma} = \left\| (I_{1,\gamma,Q_L})^\dagger \circ T \circ I_{1\beta,P_K}^* \right\|_{\mathrm{HS}(\mathcal{H}_K^\beta, \mathcal{H}_L^\gamma)}$, so that

$$\begin{aligned}
\|T\|_{\beta,\gamma}^2 &= \left\| (I_{1,\gamma,Q_L})^\dagger \circ T \circ I_{1\beta,P_K}^* \right\|_{\mathrm{HS}(\mathcal{H}_K^\beta, \mathcal{H}_L^\gamma)}^2 \\
&= \sum_{i,j=1}^{+\infty} \left\langle \rho_j^{\frac{\gamma}{2}} f_j, \left( I_{1,\gamma,Q_L}^* \right)^\dagger \circ T \circ I_{1,\beta,P_K}^* \mu_i^{\frac{\beta}{2}} e_i \right\rangle_{\mathcal{H}_L^\gamma}^2 \\
&= \sum_{i,j=1}^{+\infty} \left\langle \rho_j^{\frac{\gamma}{2}} f_j, \left( I_{1,\gamma,Q_L}^* \right)^\dagger \circ T \mu_i^{1-\frac{\beta}{2}} e_i \right\rangle_{\mathcal{H}_L^\gamma}^2 \\
&= \sum_{i,j=1}^{+\infty} \left\langle \rho_j^{\frac{\gamma}{2}} f_j, T \mu_i^{1-\frac{\beta}{2}} e_i \right\rangle_{\mathcal{H}_L}^2 \\
&= \sum_{i,j=1}^{+\infty} \left\langle \rho_j^{\frac{1}{2}} f_j, \mathcal{C}_{Q_L}^{-(1-\gamma)/2} \circ T \circ \mathcal{C}_{KK}^{(1-\beta)/2} \mu_i^{\frac{1}{2}} e_i \right\rangle_{\mathcal{H}_L}^2 \\
&= \left\| \mathcal{C}_{Q_L}^{-(1-\gamma)/2} \circ T \circ \mathcal{C}_{KK}^{(1-\beta)/2} \right\|_{\mathrm{HS}}^2
\end{aligned}$$

as desired. □

**Lemma D.2** *Under Assumption 2.2, we have*

$$\left\| (C_{KK} + \lambda I)^{-\frac{1}{2}} u \right\| \leqslant \lambda^{-\frac{\alpha}{2}} \cdot A_1 \quad P_K\text{-a.s.}$$

**Proof:** By Assumption 2.2 we have $\left\|\mathcal{C}_{KK}^{-\frac{1-\alpha}{2}} u\right\|_{\mathcal{H}_K} \leqslant A_1$, so that

$$\left\|(\mathcal{C}_{KK} + \lambda I)^{-\frac{1}{2}} u\right\| \leqslant \left\|(\mathcal{C}_{KK} + \lambda I)^{-\frac{\alpha}{2}}\right\| \cdot \left\|\mathcal{C}_{KK}^{-\frac{1-\alpha}{2}} u\right\| \leqslant \lambda^{-\frac{\alpha}{2}} \cdot A_1$$

as desired. $\qquad\square$

## D.1 CONCENTRATION INEQUALITIES

**Theorem D.1** *(Fischer & Steinwart, 2020, Theorem 27) Let $(\Omega, \mathcal{B}, P)$ be a probability space, $\mathcal{H}$ be a separable Hilbert space and $X : \Omega \to \mathrm{HS}(H; H)$ be a random variable with self-adjoint values. Furthermore, assume that $\|X\|_F \leqslant B, P - a.s.$ and $V$ be a positive semi-definite matrix with $\mathbb{E}_P\left(X^2\right) \preccurlyeq V$, i.e. $V - \mathbb{E}_P\left(X^2\right)$ is positive semi-definite. Then, for $g(V) := \log\left(2e\operatorname{tr}(V)\|V\|^{-1}\right), \tau \geqslant 1$, and $n \geqslant 1$, the following concentration inequality is satisfied*

$$P^n\left((\omega_1, \ldots, \omega_n) \in \Omega^n : \left\|\frac{1}{n}\sum_{i=1}^n X\left(\omega_i\right) - \mathbb{E}_P X(\omega)\right\| \geqslant \frac{4\tau B g(V)}{3n} + \sqrt{\frac{2\tau\|V\|g(V)}{n}}\right) \leqslant 2e^{-\tau}.$$

**Theorem D.2** *(Fischer & Steinwart, 2020, Theorem 26) Let $(\Omega, \mathcal{B}, P)$ be a probability space, $H$ be a separable Hilbert space, and $\xi : \Omega \to H$ be a random variable with*

$$\mathbb{E}_P\|\xi\|_H^m \leqslant \frac{1}{2}m!\sigma^2 L^{m-2}$$

*for all $m \geqslant 2$. Then, for $\tau \geqslant 1$ and $n \geqslant 1$, the following concentration inequality is satisfied*

$$P^n\left((\omega_1, \ldots, \omega_n) \in \Omega^n : \left\|\frac{1}{n}\sum_{i=1}^n \xi\left(\omega_i\right) - \mathbb{E}_P\xi\right\|_H^2 \geqslant 32\frac{\tau^2}{n}\left(\sigma^2 + \frac{L^2}{n}\right)\right) \leqslant 2e^{-\tau}$$

The following theorem shows that the regularized covariance $\mathcal{C}_{KK} + \lambda I$ can be estimated with small error when $\lambda$ is above a certain threshold. Although it is well-known (Fischer & Steinwart, 2020; Talwai et al., 2022), we still recall it below for completeness.

**Theorem D.3** *Recall that $\mathcal{C}_{KK} = \mathbb{E}_{P_K} u \otimes u$ and $\hat{\mathcal{C}}_{KK} = \frac{1}{N}\sum_{i=1}^N u_i \otimes u_i$ where $u_i \overset{\text{i.i.d.}}{\sim} P_K$. Suppose that Assumption 2.2 holds and $N \gtrsim A_1^2 \tau g_\lambda \lambda^{-\alpha}$, where $g_\lambda = \log\left(2e\mathcal{N}_{P_K}(\lambda)\frac{\|\mathcal{C}_{KK}\|+\lambda}{\|\mathcal{C}_{KK}\|}\right)$ and $\mathcal{N}_{P_K}(\lambda) = \operatorname{tr}\left((\mathcal{C}_{KK} + \lambda I)^{-1}\mathcal{C}_{KK}\right)$ is the effective dimension, then with probability at least $1 - e^{-\tau}$, we have*

$$\left\|(\mathcal{C}_{KK} + \lambda I)^{-\frac{1}{2}}\left(\mathcal{C}_{KK} - \hat{\mathcal{C}}_{KK}\right)(\mathcal{C}_{KK} + \lambda I)^{-\frac{1}{2}}\right\| \lesssim \sqrt{\frac{A_1^2 \tau g_\lambda}{N\lambda^\alpha}} \leqslant 0.1. \tag{28}$$

**Proof:** Let $X(u) = (\mathcal{C}_{KK} + \lambda I)^{-\frac{1}{2}} u \otimes u (\mathcal{C}_{KK} + \lambda I)^{-\frac{1}{2}}$ where $u \in \mathcal{H}_K$, then the LHS of (28) can be expressed as $\left\|\frac{1}{N}\sum_{i=1}^N X(u_i) - \mathbb{E}_{u \sim P_K} X(u)\right\|$. We hope to apply Theorem D.1 and start with verifying the assumptions.

Since $\mathbb{E}_{P_K} X = (\mathcal{C}_{KK} + \lambda I)^{-\frac{1}{2}}\mathcal{C}_{KK}(\mathcal{C}_{KK} + \lambda I)^{-\frac{1}{2}}$ and $\|X(u)\| = \|X(u)\|_F = \left\|(\mathcal{C}_{KK} + \lambda I)^{-\frac{1}{2}} u\right\|^2 \leqslant A_1^2 \left\|(\mathcal{C}_{KK} + \lambda I)^{-\frac{1-\alpha}{2}}\right\|^2 \lesssim A_1^2 \lambda^{-\alpha}$, so that there exists $V = \mathcal{O}\left(\lambda^{-\alpha}(\mathcal{C}_{KK} + \lambda I)^{-\frac{1}{2}}\mathcal{C}_{KK}(\mathcal{C}_{KK} + \lambda I)^{-\frac{1}{2}}\right)$ such that $\mathbb{E}_{P_K} X^2 \preccurlyeq V$. It's easy to see that $\|V\| \lesssim \lambda^{-\alpha}$ and $\operatorname{tr}(V) \lesssim \mathcal{N}_{P_K}(\lambda)$. The conclusion then follows from Theorem D.1 with $B = \mathcal{O}(\lambda^{-\alpha})$ and $g(V) = g_\lambda$. $\qquad\square$

**Corollary D.1** *Under the notations and assumptions of Theorem D.3, there exists a constant $C_1 > 0$ with probability $\geqslant 1 - e^{-\tau}$ we have*

$$\left\|(\mathcal{C}_{KK} + \lambda I)^{\frac{1}{2}}\left(\hat{\mathcal{C}}_{KK} + \lambda I\right)^{-1}(\mathcal{C}_{KK} + \lambda I)^{\frac{1}{2}}\right\| \leqslant C_1. \tag{29}$$

***Proof***: By Theorem D.3 we have

$$\left\| (\mathcal{C}_{KK} + \lambda I)^{\frac{1}{2}} \left( \hat{\mathcal{C}}_{KK} - \mathcal{C}_{KK} \right)^{-1} (\mathcal{C}_{KK} + \lambda I)^{\frac{1}{2}} \right\|$$
$$= \left\| \left( I - (\mathcal{C}_{KK} + \lambda I)^{-\frac{1}{2}} (\mathcal{C}_{KK} - \hat{\mathcal{C}}_{KK})(\mathcal{C}_{KK} + \lambda I)^{-\frac{1}{2}} \right)^{-1} \right\|$$
$$\leqslant 2$$

with probability $\geqslant 1 - e^{-\lambda}$, as desired. $\qquad\qquad\qquad\qquad\qquad\qquad\qquad\qquad\square$

