# OpenReview forum: "Minimax Optimal Kernel Operator Learning via Multilevel Training"
_ICLR.cc/2023/Conference — ICLR 2023 notable top 25%_

### Official Review · Reviewer_v7T2 · 2022-10-21

**Confidence:** 3
**Correctness:** 4
**Technical Novelty And Significance:** 4
**Empirical Novelty And Significance:** Not applicable
**Recommendation:** 10

**Clarity, Quality, Novelty And Reproducibility:**

The paper is written with a very clear structure, and the idea of bias and variance contours are clearly illustrated with figures. The results are novel and of high scientific quality.
Since this paper is on theoretical research, the reproducibility of theory is guaranteed by the provision of proof, and numerical reproducibility is not applicable.

**Details Of Ethics Concerns:**

The paper studies machine learning theory, and there is no ethics concern.

**Strength And Weaknesses:**

Strength
1. Clear insights provided by bias and variance contours
2. Important mathematical problem well motivated by many applications, including generative modeling, functional linear models, causal inference, multi-agent reinforcement learning, and so on.
3. Analysis achieves minimax optimal rates

Weaknesses:
No significant weakness has been identified.

**Summary Of The Paper:**

This paper studies the problem of learning trace-class operators between Sobolev reproducing kernel Hilbert spaces. Information theoretical lower bound of convergence rates are derived. The machine learning problem is characterized by bias contour and variance contour on spectral spaces. Three regularization schemes are proposed inspired by the bias and variance contours. All three schemes are proven to achieve the minimax lower bound subject to a logarithmic factor.

**Summary Of The Review:**

Inspiring characterization of bias and variance contours. Neat convergence analysis with the estimations achieving minimax optimal rate.

---

> ### Author Response · Authors · 2022-11-08
> **Response to Reviewer v7T2**
>
> Thanks Reviewer v7T2  for the review. We're happy to see the reviewer the characterization of bias and variance contours inspiring. We are very proud of this explanation and hope this can inspire researches for their future work to construct minimax optimal estimators!

---

### Official Review · Reviewer_7Ap7 · 2022-10-24

**Confidence:** 2
**Correctness:** 3
**Technical Novelty And Significance:** 3
**Empirical Novelty And Significance:** Not applicable
**Recommendation:** 8

**Clarity, Quality, Novelty And Reproducibility:**

The results obtained by the authors seem novel and high quality but the density of the paper prevents is clarity and is the main weakness.

**Strength And Weaknesses:**

### Strengths

1. The problem of deriving theoretical learning rates for Hilbert-Schmidt operators addressed by this paper is a significant topic with applications to understand performance of state-of-the-art neural operators approach for learning solutions operators of PDEs.
2. The authors derive a novel lower bound for learning linear operators between Sobolev reproducing kernel Hilbert spaces, generalizing existing works by Li et al (2022) which takes into account the output space (as opposed to prior works).
3. The lower bound on the learning rate is optimal as shown by the authors who introduces a scheme achieving a near-optimal learning rate.

### Weaknesses

1. The paper is very technical and dense which makes its understanding very difficult beyond section 1. More backbround material should be provided in section 2 on reproducing kernel Hilbert spaces and Hilbert-Schmidt operators along with concrete examples connection to operator learning associated with PDEs.
2. The lower bound provided by the authors in Theorem 3.1 is the main result for the paper but little explanation and interpretation is provided in the section and no comment on the proof techniques.
3. The multilevel kernel operator learning scheme of section 5 seems interesting (based on the theoretical results in Theorem 5.1) but I don't understand the method, which is essentially described in one sentence by Eq. (5). I would suggest rewriting the section to describe the method extensively, eventually adding a simple illustrative examples.

### Minor comments

1. End of page 3: "kernl" -> "kernels"
2. Could the authors add some interpretation on the assumptions for the kernel in section 2.2?
3. Some equations are missing punctuations, e.g. Lemma A.1, A.3...

**Summary Of The Paper:**

This paper considers the problem of deriving theoretical learning rate for learning mappings between infinite-dimensional function spaces. As noted by the authors, it is an important topic to understand how much training data is needed to approximate linear operators within a prescribed accuracy. After reviewing existing theoretical works on operator learning connected to PDEs, the authors derive a lower bound for learning linear operators between infinite-dimensional Sobolev reproducing kernel Hilbert spaces.

**Summary Of The Review:**

While the paper contains novel and significant results on the sample complexity of Hilbert-Schmidt operators, its technicality and density prevent the readability. I believe that a revised version of the paper, including the proofs in the main text and expanding on the background material and interpretation of the results, would be more suitable for a top machine learning journal like Journal of Machine Learning Research (with a longer review time allowing for a careful check of correctness) rather than a short conference paper with a long supplementary material.

--
I have updated my score following the authors' response.

---

> ### Author Response · Authors · 2022-11-08
> **Response to reviewer 7Ap7**
>
> We are grateful for Reviewer 7Ap7’s helpful comments. In the following, we address your concerns point by point.
>
> **Example for PDEs**
>
> We’ve included an example of PDEs in Example 2.1, showing that solving a linear elliptic equation is one of our examples.
>
> **Explanation of the lower bound**
>
> Sorry for the confusion. We actually add the explanation at the beginning of section 4, Remark 4.1, and Figure 2. But It’s still confusing for a non-expert to understand. Thus We add the following text as section 4.1, trying to explain the idea of our proof for non-experts. I hope this can make the reviewer understand our paper better.
>
> In this section, we provide an intuitive explanation of our lower bound (Theorem \ref{thm:lowerbound}). As the previous paragraph explains, learning an operator is equivalent to learning an "infinite" matrix with larger variance and smaller bias on the right upper corner. The core proof of this paper is considering proper bias-variance trade-off. Suppose one wants to construct an estimator with $n^\theta$ learning rate. They need to learn every spectral component below the bias counter at the level of $n^\theta$. Otherwise, the bias itself will become larger than $n^\theta$. At the same time, they still need not learn any spectral component above the variance counter at the level of $n^\theta$. Otherwise, the variance itself will become larger than $n^\theta$. Thus the variance counter at the level of $n^\theta$ should always be above the bias counter at the level of $n^\theta$ (Figure 2). Depending on the hyperparameters, there are two different ways to achieve this goal, as shown in Figure 2. Each situation is mapped to the rate depending only on the input space, and the rate depends only on the output space. In Section 5, we demonstrated how a multilevel algorithm could be used to satisfy this requirement.
>
> Our lower bound is proved using  Fano’s method, the standard technique the authors know can imply information-theoretical lower bounds. We add this before theorem 3.1.
>
>
> **Regards the multilevel training**
>
> Actually, another interesting thing about our estimator is that it aligns with empirical use. [1,2] proposed multilevel training for the neural network-based operator learning scheme. Our multilevel algorithm first applies the regression algorithm on low-frequency projections of the output samples with small regularization and then successively fine-tunes the regression model on high-frequency projections of the output samples with stronger regularization. We add this to a remark after theorem 5.1. We hope this can help the audience understand our paper better.
>
>
> [1] Li Z, Fan Y, Ying L. Multilevel fine-tuning: Closing generalization gaps in approximation of solution maps under a limited budget for training data. Multiscale Modeling & Simulation Lye
>
> [2] K O, Mishra S, Molinaro R. A multi-level procedure for enhancing accuracy of machine learning algorithms. European Journal of Applied Mathematics, 2021, 32(3): 436-469.
>
> We hope this can help the reviewer to understand our paper better. We’re also very curious to hear from the reviewer whether our modification is clear enough to understand. If the concerns are addressed, we hope the reviewer can reevaluate our paper. If this still confuses you, we are always willing to make further revisions before the discussion deadline.

---

> > ### Author Response · Authors · 2022-11-15
> > **Follow up**
> >
> > We're curious about what the reviewer thinks about our revision. Has our revision have resolves the reviewer's concern? Can our revision help the reviewer to reevaluate our contribution? The discussion deadline is approaching in two days. If the reviewer still has concerns about our presentation, we still have time to do another round of revision.
> >
> > We are looking forward to hearing from the reviewer.

---

> > > ### Comment · Reviewer_7Ap7 · 2022-11-16
> > > **Response to authors**
> > >
> > > I thank the authors for the response which addresses my main comments. While I am still concerned about the density of the paper for ICLR, I still believe that it is a great paper that deserves to be published and hence raise my score to recommend acceptance.
> > >
> > > #### Minor comments to be addressed before publication:
> > > - There are a few typos remaining in the paper, e.g. "divided into defferent", "Our multilevel algorithm first apply" p.9 so I would recommend running a spell checker to correct those.
> > > - A longer discussion of the multilevel algorithm in Remark 5.1 would be great to discuss potential practical outcomes of Theorem 5.1.

---

> > > > ### Author Response · Authors · 2022-11-16
> > > > **Thank you and sorry for the typo**
> > > >
> > > > Thanks for the review, which makes our paper more suitable for a wider audience. Sorry for the typos. We will check our paper carefully these days and revise it before the discussion deadline.

---

### Official Review · Reviewer_EtQp · 2022-10-24

**Confidence:** 2
**Correctness:** 4
**Technical Novelty And Significance:** 3
**Empirical Novelty And Significance:** Not applicable
**Recommendation:** 8

**Clarity, Quality, Novelty And Reproducibility:**

As far as I can tell the paper is clear, and the results would appear to be novel.  I did struggle with some of the mathematics in the paper but I assume this has more to do with unfamiliarity on my part than any problems with the presentation itself.

**Strength And Weaknesses:**

The paper is soundly written and to the best of my knowledge the theoretical contributions are novel and provide insight into the problem.  Examples are given to motivate the mathematical development and there is commentary describing salient points.

My one complaint is the lack of any summary of proofs in the body of the paper.  While obviously it is not practical to include the actual proofs, it is useful to include some indication of the important steps or interesting methods required in the body of the paper.

**Summary Of The Paper:**

The paper analyses the problem of constructing maps netween infinite dim function spaces.  The paper claims 3 contributions:

1. An information-theoretic lower bound of learning a linear operator between two infinite-dimensional Sobolev RKHSs consisting of two polynomial rates, the first of which depends on the input space and is consistent with known results, the second of which depends on the output space and is novel.

2. A study of the shape of regularization needed to obtain the optimal learning rate.

3. A setting where a multilevel training procedure is needed to achieve the optimal learning rate, which is distinct from the finite-dimensional case where a single-level estimator suffices.

**Summary Of The Review:**

The paper presents novel results in what appears to be a thorough and informative manner.

---

> ### Author Response · Authors · 2022-11-08
> **Response to Reviewer EtQp**
>
> Thanks, reviewer EtQp for your helpful comments. We're happy to see the reviewer find our paper novel and we're sorry that lack of any summary of proof in the main text. We’ve revised our paper according to the reviewer’s feedback as follows.
>
> **Regards the summary of the proofs**
>
> Sorry for the confusion. We actually add the explanation of our proof in the beginning of section 4, Remark 4.1, and Figure 2. But It’s still confusing for a non-expert to understand. Thus We add the following text as section 4.1, trying to explain the idea of our proof for non-experts. Hope this can make the reviewer understand our paper better.
>
> In this section, we provide an intuitive explanation of our lower bound (Theorem \ref{thm:lowerbound}). As the previous paragraph explains, learning an operator is equivalent to learning an "infinite" matrix with larger variance and smaller bias on the right upper corner. The core proof of this paper is considering proper bias-variance trade-off. Suppose one wants to construct an estimator with $n^\theta$ learning rate. They need to learn every spectral component below the bias counter at the level of $n^\theta$. Otherwise, the bias itself will become larger than $n^\theta$. At the same time, they still need not learn any spectral component above the variance counter at the level of $n^\theta$. Otherwise, the variance itself will become larger than $n^\theta$. Thus the variance counter at the level of $n^\theta$ should always be above the bias counter at the level of $n^\theta$ (Figure 2). Depending on the hyperparameters, there are two different ways to achieve this goal, as shown in Figure 2. Each situation is mapped to the rate depending only on the input space, and the rate depends only on the output space. In Section 5, we demonstrated how a multilevel algorithm could be used to satisfy this requirement.

---

> > ### Comment · Reviewer_EtQp · 2022-11-17
> > **Response to rebuttal**
> >
> > Thank you for responding and adjusting the paper to include a summary of the proof.  While I still don't feel I fully appreciate this paper, based on the responses from other reviewers I will keep my recommendation as recommend acceptance.

---

### Official Review · Reviewer_Zt4i · 2022-10-24

**Confidence:** 3
**Correctness:** 4
**Technical Novelty And Significance:** 3
**Empirical Novelty And Significance:** Not applicable
**Recommendation:** 8

**Clarity, Quality, Novelty And Reproducibility:**

The paper is well-written with a sufficient literature review. The results are novel, significant, and of broad interest.

**Strength And Weaknesses:**

$\textbf{Strength}$

1. This is a novel contribution that provides insights into the understanding of operator learning between two infinite-dimensional spaces, a topic with various applications in scientific computing, machine learning, and statistics.

2. The minimax lower bound for learning a linear operator is interesting and novel, which characterized the difficulty of this problem.

3. The proposed multi-level kernel operator learning algorithm also makes a solid contribution and the insights are valuable for the design of other operator learning algorithms.

4. The mathematical analysis of this paper is rigorous and sound, which extends the previous results for learning differential operators and conditional mean embedding.

$\textbf{Concerns}$

1. Can the analysis (or lower bound) be extended to nonlinear operator learning?

2. The definitions of $\hat{C}_{LK}$ in (3) and (5) are not given. It is also a little confusing how $\hat{A}$ can be calculated in practice as $\rho_j$ and $f_j$ are not easy to obtain.

**Summary Of The Paper:**

This paper considers the problem of learning a linear operator between two infinite-dimensional Sobolev reproducing kernel Hilbert spaces, which includes learning differential operators and condition mean embedding as special examples. A novel information-theoretical lower bound is derived, which implies that the minimax learning rate depends on both the smoothness of input and output spaces. This paper proposes a multi-level kernel operator learning algorithm that can achieve the optimal learning rate.

**Summary Of The Review:**

The paper studies the optimal convergence rate of learning a linear operator between two infinite-dimensional spaces, which should be of interest to a broader audience in the field of machine learning and scientific computing.

---

> ### Author Response · Authors · 2022-11-08
> **Response to reviewer Zt4i**
>
> We are grateful for Reviewer Zt4i’s helpful comments. In the following, we answer your questions point by point as follows.
>
> **Regards nonlinear operator learning**
>
> The development of functional nonparametric regression has been hindered by a theoretical barrier. Non-parametric functional data analysis always surfers from converge rate slower than any polynomial rate [1] and is linked to the small ball probability problem for probability distribution in infinite dimensional space [2]. The existing literature on nonlinear operator learning [3] also suffers from the slower than the polynomial rate. Making proper assumptions to break this barrier is still an open problem, as far as the authors know.
>
> [1] Lin Z, Yao F. Functional regression on the manifold with contamination[J]. Biometrika, 2021, 108(1): 167-181.
>
> [2]  MAS, A. (2012). Lower bound in regression for functional data by representation of small ball probabilities. Electronic Journal of Statistics 6 1745–1778
>
> [3] Liu H, Yang H, Chen M, et al. Deep nonparametric estimation of operators between infinite dimensional spaces[J]. arXiv preprint arXiv:2201.00217, 2022.
>
>
> **Definition of $\hat C_{LK}$**
>
> $\hat C_{LK}$ is the empirical version is $C_{LK}$ and is defined as $\hat C_{LK} = \frac{1}{N}\sum_{i=1}^N v_i  \otimes u_i$. Here $(u_i,v_i),1\leq i\leq N$ is the data. We’ve added the notation in the revision.
>
>
> **Regards $\rho_j$ and $f_j$**
>
> Yes, $\rho_j$ and $f_j$ is hard to compute in general. In our paper, we set the output covariance $C_{Q_L}$ is defined by a user-specified distribution $Q_L$ following the setting in [1] for PDE learning. Thus it is often possible to obtain $f_j$ explicitly. For instance, in Example 2.1 we have $f_j(z) = \sqrt{2}\sin(j\pi z)$ [1,Sec.4.1] (Fourier basis) and $\rho_j \propto j^{-2m}$. We admit this assumption is not satisfactory for kernel mean embedding where  $\rho_j$ and $f_j$ should be estimated from data. However, doing this will introduce another statistical error which is not easy to bound and make the analysis nasty. Thus, we leave it as future work.
>
> [1] de Hoop, Maarten V., et al. "Convergence rates for learning linear operators from noisy data." arXiv preprint arXiv:2108.12515 (2021).

---

> > ### Comment · Reviewer_Zt4i · 2022-11-18
> > **Response to authors**
> >
> > I would like to thank the authors for their responses and clarifications, which have addressed all my concerns. I maintain my positive appraisal.

---

### Official Review · Reviewer_B1fn · 2022-10-25

**Confidence:** 3
**Correctness:** 4
**Technical Novelty And Significance:** 3
**Empirical Novelty And Significance:** Not applicable
**Recommendation:** 10

**Clarity, Quality, Novelty And Reproducibility:**

Clarity

Overall, the paper was written in a very clear manner for kernel experts. It would be nice to at least point to other works that interpret the various abstract assumptions of Fischer and Steinwart’s paradigm in a way that more readers can understand.

The remarks do a great job explaining the similarities and differences with prior work. Two additional references which I would like to be included in the “learning with kernel” paragraph of the related work are
1. “A measure-theoretic approach to kernel conditional mean embeddings” (NeurIPS 2020), which was a relatively early work about learning rates, though not in Sobolev norm
2. “Kernel methods for causal functions” (arXiv 2022), which obtains optimal rates in Sobolev norm but only considers beta=1 and gamma=1
Neither of these references diminish of the contributions of this work in any way, but would make the related work section more complete.

Quality

I only skimmed the proofs due to time constraints. The main results and main steps in the proof seemed reasonable.

Novelty

By studying the harder learning scenario in which both the input and output RKHSs are mis-specified, the authors break new ground. The second rate in the bound, based on the output RKHS, seems to be interesting and novel.

Reproducibility

Given the proposal of a new multilevel estimator, I was a bit surprised that there were no simulations. The works in this literature that study well known estimators sometimes skip the simulations, but this paper seems to propose a new estimator and to claim that in certain “hard” scenarios it is preferable. It raises the question of how difficult the new multilevel estimator is to implement.

**Strength And Weaknesses:**

See below

**Summary Of The Paper:**

The authors study the problem of learning a linear operator from one RKHS to another RKHS (hereafter from an input RKHS to an output RKHS). In this literature, it is common to study a certain kind of mis-specification: what if the ground truth belongs to an interpolation space between the RKHS and L2. Whereas previous works on the problem consider the mis-specification with respect to the input RKHS, this work considers mis-specification with respect to both the input RKHS and the output RKHS, which is a generalization. The authors characterize the learning lower bound in generalized Hilbert-Schmidt norm and analyze a new multilevel estimator that attains it.

**Summary Of The Review:**

This paper joins a mature literature, but I think that its study of the output RKHS mis-specification is significant. The lack of simulations raises the question of how feasible the multilevel estimator actually is. If the authors implement the minor improvements suggested above, I will raise the score.

---

> ### Author Response · Authors · 2022-11-08
> **Response to reviewer B1fn**
>
> We are grateful for Reviewer B1fn’s helpful comments. In the following, we address your concerns point by point. Hope our explanation on the empirical use of our estimator can help the reviewer to understand our motivation.
>
> Sorry for missing the reference [1,2].  Definitely, they are related to our paper, and thanks to the reviewer for distinguishing the difference between our paper and theirs. We’ve updated our paper and added them as our reference.
>
> [1] A measure-theoretic approach to kernel conditional mean embeddings
> [2] Kernel methods for causal functions
>
>
> **Regards Empirical use**
>
> Actually, another interesting thing about our estimator is it aligns with empirical use. [1,2] proposed multilevel training for the neural network-based operator learning scheme, and [3,4,5] proposed a multi-level training method for learning the green function.  All these papers are empirical motivations for our paper.
>
> Our multilevel algorithm first applies the regression algorithm on low-frequency projections of the output samples with small regularization and then successively fine-tunes the regression model on high-frequency projections of the output samples with stronger regularization. We add this to a remark after theorem 5.1. We hope this can help the audience understand our paper better. We are also curious if our answer addresses your concern. If the reviewer still has questions, we’d love to have a further explanation.
>
> Sorry again for note make this clear in our previous version of main text.
>
>
> [1] Li Z, Fan Y, Ying L. Multilevel fine-tuning: Closing generalization gaps in approximation of solution maps under a limited budget for training data. Multiscale Modeling & Simulation Lye
>
> [2] K O, Mishra S, Molinaro R. A multi-level procedure for enhancing accuracy of machine learning algorithms. European Journal of Applied Mathematics, 2021, 32(3): 436-469.
>
> [3] Lin L, Lu J, Ying L. Fast construction of hierarchical matrix representation from matrix–vector multiplication. Journal of Computational Physics, 2011, 230(10): 4071-4087.
>
> [4] Schäfer F, Owhadi H. Sparse recovery of elliptic solvers from matrix-vector products. arXiv preprint arXiv:2110.05351, 2021.
>
> [5] Boullé N, Kim S, Shi T, et al. Learning green’s functions associated with time-dependent partial differential equations. Journal of Machine Learning Research, 2022, 23(218): 1-34.

---

> > ### Comment · Reviewer_B1fn · 2022-11-18
> > **I am raising the score**
> >
> > Thank you for these improvements and further explanations.
> >
> > In particular, I find the significance of the theoretical contribution to be enhanced by pointing out that the multilevel algorithm in Section 5 is closely related to others that are in use. After looking at the cited works in the new remark, I am not sure that it perfectly coincides, but they are reasonably close and the theoretical contribution of this paper is strong enough to defer an empirical comparison to future work
> >
> > Overall, I congratulate the authors on a rigorous paper

---

### Decision · Program_Chairs · 2023-01-20

**Decision:**

Accept: notable-top-25%

**Justification For Why Not Higher Score:**



**Justification For Why Not Lower Score:**



**Metareview: Summary, Strengths And Weaknesses:**

In this paper the authors focus on the learning problem of Hilbert-Schmidt operators between infinite-dimensional Hilbert spaces (more particularly, between interpolation Sobolev spaces). The motivation comes from learning solution operators of partial differential equations and conditional mean embedding. They present an information-theoretic lower bound for the convergence rate (using the Fano method; Theorem 3.1), and inspired by multilevel Monte-Carlo (MLC) techniques, they present an MLC-type scheme (the multi-level kernel operator learning algorithm) which is proved to be (nearly, modulo logarithmic factors) minimax-optimal as shown in Theorem 5.1.

Kernel methods are on the forefront of data science, designing novel theoretically-grounded methods is of clear interest to the machine learning and ICLR community. The authors achieve this goal in a well-written, rigorous and novel work as it was assessed by the reviewer.

**Note From Pc:**

if the above contains the word "oral" or "spotlight" please see: "oral" presentation means -> notable-top-5% and "spotlight" means -> notable-top-25%. As stated in our emails, we are disassociating presentation type from AC recommendations